# Analysis of Cultured Gut Microbiota Using MALDI-TOF MS in COVID-19 Patients from Serbia during the Predominance of the SARS-CoV-2 Omicron Variant

**DOI:** 10.3390/microorganisms12091800

**Published:** 2024-08-30

**Authors:** Aleksandra Patić, Gordana Kovačević, Vladimir Vuković, Ivana Hrnjaković Cvjetković, Mioljub Ristić, Biljana Milosavljević, Deana Medić, Milan Djilas, Jelena Radovanov, Aleksandra Kovačević, Tatjana Pustahija, Dragana Balać, Vladimir Petrović

**Affiliations:** 1Institute of Public Health of Vojvodina, 21000 Novi Sad, Serbia; aleksandra.patic@mf.uns.ac.rs (A.P.); vladimir.vukovic@mf.uns.ac.rs (V.V.); ivana.hrnjakovic-cvjetkovic@mf.uns.ac.rs (I.H.C.); mioljub.ristic@mf.uns.ac.rs (M.R.); biljana.milosavljevic@izjzv.org.rs (B.M.); deana.medic@mf.uns.ac.rs (D.M.); milan.djilas@izjzv.org.rs (M.D.); jelena.radovanov@izjzv.org.rs (J.R.); tatjana.pustahija@mf.uns.ac.rs (T.P.); tistudb@gmail.com (D.B.); vladimir.petrovic@mf.uns.ac.rs (V.P.); 2Department of Microbiology with Parasitology and Immunology, Faculty of Medicine, University of Novi Sad, 21000 Novi Sad, Serbia; 3Faculty of Medicine, University of Novi Sad, 21000 Novi Sad, Serbia; 4Department of Epidemiology, Faculty of Medicine, University of Novi Sad, 21000 Novi Sad, Serbia; 5Clinic of Nephrology and Clinical Immunology, University Clinical Center of Vojvodina, 21000 Novi Sad, Serbia; sanja.kovacevic021@gmail.com; 6Department of Hygiene, Faculty of Medicine, University of Novi Sad, 21000 Novi Sad, Serbia

**Keywords:** COVID-19, SARS-CoV-2, MALDI-TOF MS, gut microbiota

## Abstract

The currently dominant SARS-CoV-2 omicron variant, while causing mild respiratory symptoms, exhibits high transmissibility, drug resistance, and immune evasion. We investigated whether the presence of the SARS-CoV-2 affected the dynamics of fecal microbial composition isolated in culture in moderate COVID-19 patients. Blood, stool, and medical records were collected from 50 patients with confirmed SARS-CoV-2 infection. Two samples were taken per patient, at disease onset (within 5 days) and after symptom resolution (30–35 days). The part of the gut microbiota identifiable using MALDI-TOF MS was analyzed, and inflammatory cytokines and blood markers were measured in serum. The analysis identified 566 isolates at the species level, including 83 bacterial and 9 fungal species. Our findings indicate a change in the gut microbiota composition isolated in culture during the initial phase of infection, characterized by the proliferation of opportunistic bacteria such as *Enterococcus* spp. and *Citrobacter* spp., at the expense of beneficial commensal bacteria from the genus *Bacillus* and *Lactobacillus*. Additionally, the enrichment of fungal pathogens in fecal samples collected 30 days after the cessation of disease symptoms might suggest a prolonged disruption of the gut microbiota even after the resolution of COVID-19 symptoms. This study contributes to a growing body of evidence on the systemic effects of SARS-CoV-2 and highlights the importance of considering gastrointestinal involvement in the management and treatment of COVID-19.

## 1. Introduction

The term gut microbiota refers to the collection of resident microorganisms that inhabit the gastrointestinal tract (GIT), including bacteria, fungi, viruses, phages, and archaea [1,2]. Various factors, such as nutrition, genetics, environment, and lifestyle, influence the composition of gut microbiota [3,4]. Bacteria are essential components of this community due to their numerous functions, including food fermentation, protection against pathogens, immune response stimulation, and vitamin production [5,6]. Generally, bacteria inhabiting the gastrointestinal tract are categorized into six phyla: *Firmicutes*, *Bacteroidetes*, *Actinobacteria*, *Proteobacteria*, *Fusobacteria*, and *Verrucomicrobia*, with *Firmicutes* and *Bacteroidetes* being the dominant taxa [4]. The gut microbiome plays an important role in immune system regulation, and alterations in its composition have been observed in various infectious diseases [7,8]. Different bacterial and viral infections, including SARS-CoV-2, have previously been associated with persistent gastrointestinal symptoms and post-infectious irritable bowel syndrome [9]. A healthy gut microbiota helps protect against the colonization of harmful pathogens by competing for resources and producing antimicrobial substances [10]. This competitive exclusion helps maintain a balanced and protective environment in the gut. Understanding the intricate relationship between the gut microbiota and human health is an active area of research, and ongoing studies continue to uncover new insights into the role of these microorganisms in maintaining overall human health [11,12].

The pathogenesis of coronavirus disease (COVID-19) is a consequence of the severe acute respiratory syndrome coronavirus 2 (SARS-CoV-2) virus’s tropism, which is determined by the distribution of the angiotensin-converting enzyme 2 (ACE2) receptor. In humans, the ACE2 receptor is present in the nose, lungs, kidneys, heart, and gastrointestinal tract. This distribution enables SARS-CoV-2 to infect not only human respiratory epithelial cells but also the cells of the gastrointestinal (GI) system [13,14]. In addition, several studies conducted in different regions have suggested that the gut microbiota may contribute to the pathogenesis of COVID-19 and the disease outcomes [2,15]. It has also been confirmed that SARS-CoV-2 infection is associated with an altered composition of the intestinal microbiota and correlated with inflammatory and immune responses [10,13]. However, the largest number of clinical and preclinical studies on the impact of SARS-CoV-2 on the gut microbiota have focused on earlier SARS-CoV-2 variants that caused severe forms of the disease [16]. Since the start of the COVID-19 pandemic, the World Health Organization (WHO) has designated several strains as variants of concern (VOCs). These include strains B.1.1.7 (alpha), B.1.351 (beta), P.1 (gamma), B.1.617.2 (delta), and B.1.1.529 (omicron) [17]. Compared to previous variants, omicron is characterized by increased transmissibility and milder forms of the disease, accompanied by a reduced rate of hospitalization and a shorter recovery time [18]. The virus has continued to evolve and accumulate mutations, particularly in the receptor-binding domain of the spike protein. This has led to the simultaneous rise of multiple omicron descendants, sharing common mutations that enhance the virus’s ability to evade neutralizing antibodies. Omicron subvariants currently circulating globally include JN.1, KP.2, KP.3, and KP.1 [19].

Given the global predominance of the omicron variant and its subvariants and the lack of research on the impact of SARS-CoV-2 on intestinal microflora in Serbia and the region, our study aims to explore the dynamics of microbial composition isolated in culture in SARS-CoV-2-positive patients with moderate clinical symptoms during their active disease and in the convalescence period. We seek to analyze the microbial composition isolated in culture at the onset of illness and one month after recovery and to determine whether any potential disruptions in the gut microbiota composition resolve after SARS-CoV-2 viral clearance and whether they are associated with disease severity in patients with COVID-19. Additionally, we investigated whether alterations in the gut microbiota composition and diversity correlate with biomarkers of inflammatory response in COVID-19 patients.

## 2. Materials and Methods

### 2.1. Sample Collection

This prospective cohort study included 50 adult patients with a laboratory-confirmed diagnosis of COVID-19, defined as a positive SARS-CoV-2 result from nose/throat swabs validated by either the RT-PCR method or a rapid antigen test. The research was conducted at the Institute of Public Health of Vojvodina, Novi Sad, Serbia, from July 2022 to August 2023. Within five days after a confirmed positive result, initial stool and blood samples were collected from study participants, followed by additional stool and blood samples 30–35 days after the cessation of disease symptoms. At the initial sampling, participants provided data on their socio-demographic characteristics, current illness symptoms, history of COVID-19, COVID-19 vaccination status, and comorbidities. The progression or regression of disease symptoms, along with details on therapy and treatment, was monitored every three days until symptom resolution.

The study included only individuals who had not used antibiotics and probiotics for at least a month prior to enrollment, were not undergoing chemotherapy or radiation therapy, were not immunocompromised (e.g., HIV/AIDS, transplant, cancer, etc.), and did not have a history of inflammatory bowel disease. Pregnant women and children under 18 years of age were not included in the research. An overview of the study procedure and general sampling process is presented in Figure 1.

Informed written consent was obtained from each participant. The study protocol was approved by the Ethics Committee of the Institute for Public Health of Vojvodina (Decision number: 01-991/1-1 of 27 June 2022).

### 2.2. Detection of SARS-CoV-2 RNA in Stool Samples

Study participants received detailed instructions for the proper collection and transport of fecal samples. Immediately upon arrival at the laboratory, fecal samples were processed to create a 10% suspension by homogenizing 1 g of fecal material in 9 mL of sterile balanced salt solution (0.9% NaCl) while the remainder was sent for microbiological analysis. Approximately 150 µL of this prepared suspension served as the starting material for the extraction of viral nucleic acid. If fecal suspensions could not be immediately subjected to nucleic acid isolation, they were stored at −80 °C. Viral RNA was extracted using the Ribovirus extraction kit (Sacace, Biotechnologies, Como, Italy) following the manufacturer’s instructions. The isolated RNA was stored at −80 °C until amplification. Detection of SARS-CoV-2 RNA was performed using the commercially available GeneProof SARS-CoV-2 PCR kit (GeneProof a.s., Brno, Czech Republic).

### 2.3. Bacterial Identification and Characterization

Identification and characterization of bacterial species within fecal samples were performed using matrix-assisted laser desorption/ionization time-of-flight mass spectrometry (MALDI-TOF MS) technology, a soft ionization technique achieved by mixing a matrix with bacterial colonies on metal plates. Mass spectra were acquired using a Microflex BioTyper spectrometer (Bruker Daltonics, Billerica, MA, USA) equipped with a nitrogen laser and controlled by Flex Control software ver. 3.4, Build 135.10 (Bruker Daltonics). Spectra were generated by measuring the time of flight of ions through the tube to the detector. Spectral comparison with the data from the available defined database was automatically performed and permitted identification.

A total of 100 stool samples (comprising both the first and second samples from COVID-19 patients) were prepared for analysis. Stool specimens were inoculated onto several nutrient media, including Columbia Blood agar, Endo agar, Salmonella Shigella (SS) agar, Schaedler agar, and Sabouraud agar. Following the culture method of seeding samples on nutrient media and subsequent incubation, the most numerous isolated colonies from each nutrient medium were selected and inoculated onto new plates to obtain pure cultures. For the direct colony method, bacteria were applied as thin films to 96-spot, polished, stainless steel target plates using sterile toothpicks (Bruker Daltonik GmbH, Leipzig, Germany). The bacteria were then left to dry at room temperature for 1 min. Subsequently, 1 μL of the matrix solution, containing a saturated α-cyano-4-hydrocinnamic acid (Bruker Daltonik) in 50% acetonitrile (Sigma-Aldritch, Burlington, MA, USA) and 2.5% trifluoroacetic acid (Sigma-Aldritch), was applied to the samples and co-crystallized with them at room temperature for 10 min. Spectra in the mass range of 2 to 20 kDa were collected using the Auto Execute option by accumulating 240 laser shots acquired at 30–40% of maximum laser power.

### 2.4. Detection of SARS-CoV-2 IgG Antibodies and Inflammatory Biomarkers

All serum samples collected from project participants were tested to determine the presence of anti-SARS-CoV-2 IgG antibodies. The analysis was conducted on serum samples with volumes ranging from 0.5 to 1 mL utilizing the “LIAISON^®^ SARS-CoV-2 TrimericS IgG” test developed by LIASON DiaSorin (DiaSorin SpA, Saluggia, VC, Italy). The tests were executed in a fully automated manner on the LIAISON^®^ XL (DiaSorin SpA, Saluggia, VC, Italy) Analyzer. Values greater than or equal to 13.0 AU/mL were interpreted as positive.

An immunoturbidimetric method was used to determine human C-reactive protein (CRP) levels in serum. Values between 1 and 5 mg/L were considered normal. Levels of pro-inflammatory and anti-inflammatory cytokines were quantified using commercially available enzyme-linked immunosorbent assay (ELISA) kits. Specifically, immunoassays for human interleukin 1 alpha (IL-1α), human interleukin 6 (IL-6), and human interleukin (IL-10), manufactured by Elabscience Bionovation Inc. (Houston, TX, USA), were employed. Absorbance readings at 450 nm were performed using an automated ELISA reader for microtiter plates. The concentrations of the target analytes were determined based on the standard curve and expressed in pg/mL. Using CurveExpert Professional 2.7.3 software, the target concentrations of the samples were interpolated from the standard curve.

Since the manufacturer, Elabscience, did not specify the reference range for pro- and anti-inflammatory cytokines, reference values were taken from the relevant scientific literature. The reference range in healthy individuals is approximately 5.186 pg/mL for IL-6 [20]; 4.8–9.8 pg/mL for IL-10 [21]; and 0–5 pg/mL for IL-1α [22].

### 2.5. Statistical Analysis

All collected data were inserted into a specially created database and used in the analyses. Results were presented as means ± standard deviation (SD). The chi-square test was applied to compare the frequencies of different bacterial taxa. For data following a normal distribution, paired *t*-tests were utilized. Categorical data were analyzed using the McNemar test. Outcomes were compared, including changes in the presence of specific bacterial taxa at the onset of infection and after the resolution of symptoms. The Pearson correlation coefficient (r) was computed to assess the strength and direction of correlations. Statistical analyses and data visualization were performed using GraphPad Prism (8.0.1 software, GraphPad software Corporation, San Diego, CA, USA), Microsoft Power BI, and Microsoft Excel 2016 (Microsoft Office, Redmond, WA, USA, Professional Plus 2016). A *p*-value of <0.05 was considered statistically significant.

## 3. Results

### 3.1. Analysis of SARS-CoV-2 RNA in Stool Samples

A cohort of 50 patients with SARS-CoV-2 infection included 22 men and 28 women. The mean age of the cohort was 50.5 years. The clinical presentation varied, with the majority (64%, 32/50) presenting with a moderate illness, followed by 24% (12/50) with a moderately severe profile. A severe illness was noted in 8% (4/50) of the cases, while 4% (2/50) were asymptomatic. Within five days of symptom onset, a total of 66% (33/50) of patients excreted SARS-CoV-2 RNA in their first stool sample. Notably, only one patient (2%) continued to shed viral RNA 30–35 days after the cessation of disease symptoms. About 42 (84%) of the study participants had received at least two doses of the COVID-19 vaccine. Approximately 47 (94%) of the study participants had SARS-CoV-2 IgG antibodies at the beginning of the infection (first blood sample), while the second measurement after 30–35 days showed that all participants had SARS-CoV-2 IgG antibodies. Half of the study participants reported gastrointestinal (GI) complaints in addition to other symptoms. Table 1 compares detailed characteristics of individuals with and without SARS-CoV-2 RNA fecal shedding in their first stool sample, including data on their sex, age group, COVID-19 severity, duration of symptoms, gastrointestinal symptoms, prior infection status, and vaccination status. As demonstrated in Table 1, the presence of SARS-CoV-2 RNA was not associated with the explored demographic and clinical characteristics of the patients.

### 3.2. Identification and Analysis of Bacterial Taxa in Fecal Samples from COVID-19 Patients

A total of 100 fecal samples were analyzed using MALDI-TOF mass spectrometry. Of these, 50 were collected during the initial phase of COVID-19 (within 5 days of a laboratory-confirmed positive COVID-19 test result) and 50 were collected 30 to 35 days after the cessation of disease symptoms. Extensive analysis identified 566 isolates at the species level (Appendix A). Using various cultivation techniques, 83 bacterial species and nine fungal species were identified. In fecal samples collected during the initial phase of the disease, a total of 272 isolates were identified. Subsequent sampling after 30–35 days revealed an increase in species diversity, with 294 isolates identified (Figure 2A).

Among the 83 identified bacterial species, 13 (15.7%) were observed only once, 45 (54.2%) were detected in two to five different samples, 14 species (16.9%) were found in six to ten different samples, and 11 species (13.2%) were identified in more than ten distinct samples. At the phylum level, *Firmicutes*, *Proteobacteria*, and *Actinobacteria* were the most dominant, while *Bacteroidetes* were identified in a limited number of isolates. These phyla were present in both sample groups with no significant changes in relative abundance based on the χ^2^-test (Figure 2B). *Enterococcus faecium* was the most commonly identified bacterial species, with 71 isolates, followed by *Escherichia coli*, with 47 isolates, *Hafnia alvei*, with 40 isolates, *Citrobacter braakii*, with 18 isolates, and both *Citrobacter freundii* and *Enterococcus faecalis*, with 17 isolates each. The analysis of bacterial genera showed an abundance of *Enterococcus*, which was present in 21.3% of the first fecal samples and 15.6% of the second samples. Other genera identified in notable quantities in the first samples included *Bacillus* (9.1%), *Escherichia* (8.4%), *Citrobacter* (7.7%), *Hafnia* (7.3%), and *Pseudomonas* (6.9%). Additionally, in the first samples, fungi were found in 2.9% of isolates. In the second samples, collected 30 to 35 days after the cessation of disease symptoms, the genera *Bacillus* (11.3%), *Pseudomonas* (8.1%), *Escherichia* (8.1%), *Hafnia* (6.8%), and *Citrobacter* (6.1%), as well as fungi (6.1%), were found. These findings were visually represented using percentages in a stacked bar plot (Figure 2C).

To assess the diversity of gut community species at the onset of illness and 30–35 days after the cessation of disease symptoms, Shannon’s diversity index (Figure 2D) and Simpson’s index (Figure 2E) were applied. Although both indices demonstrated species abundance in the community, the observed differences were not statistically significant based on the t-test (Simpson, *p* = 0.906; Shannon, *p* = 0.620).

### 3.3. Analysis of Commensal and Opportunistic Bacterial Taxa Components

Considering pathogenic potential, host interaction, environmental conditions, host status, and genetic and phenotypic characteristics, we classified the identified bacterial species and genera into opportunistic and commensal categories. We examined the differences in their presence between the initial fecal samples collected within 5 days of a laboratory-confirmed positive COVID-19 test and those collected 30–35 days after the cessation of disease symptoms. Our results indicated that opportunistic taxa (including genera such as *Enterococcus*, *Pseudomonas*, *Klebsiella*, *Citrobacter*, *Enterobacter*, *Proteus Moellerella*, *Yersinia* and *Serratia*), along with fungal pathogens, were more prevalent than commensal species at the onset of the disease (60.3% vs. 39.7%). However, in samples collected 30–35 days after the cessation of disease symptoms, this difference was no longer observed (49.2% vs. 50.8%).

Further, we focused on the presence of bacterial genera to assess whether they changed significantly over time using McNemar’s test with a significance level of 0.05 (Table 2). Since we performed multiple comparisons (10 tests), Bonferroni correction was applied to adjust the significance threshold. The adjusted significance level was α′ = 0.005. *Enterococcus faecium* emerged as the most frequently detected species among COVID-19 patients, along with the genus *Enterococcus*. Despite this, McNemar’s test indicated no significant change in the number of *Enterococcus* detections from the beginning of the disease to 30–35 days after the cessation of disease symptoms, likely due to the high number of patients with persistent *Enterococcus* presence. Moreover, the presence of opportunistic *Proteobacteria* belonging to the order *Eubacteriales* (including genera such as *Klebsiella*, *Citrobacter*, *Enterobacter*, *Proteus*, *Moellerella*, *Yersinia*, and *Serratia*) decreased during the follow-up period. However, this change was not statistically significant after the Bonferroni correction. Additionally, while the initial analysis suggested a higher presence of fungi in samples collected 30–35 days after the cessation of disease symptoms, this finding did not remain statistically significant after the correction for multiple comparisons.

The total number of commensal species detected at the onset of the disease (39.7%) increased in samples collected 30–35 days after the cessation of disease symptoms (50.8%). However, there were notable shifts in their distribution. Members of the order *Bacillales* (including the genera *Bacillus*, *Sporosarcina*, *Lysinibacillus*, *Psychrobacillus*, *Listeria*, *Paenibacillus*, and *Peribacillus*) showed a depletion. Conversely, species from the order *Lactobacillales* (including *Lactobacillus*, *Carnobacterium*, *Filifactor Limosilactobacillus*, *Streptococcus*, *Ligilactobacillus*, and *Tissierella*) showed an enrichment in the second sample compared to the first. Initially, some *p*-values related to changes in the presence of species from *Bacillales* and *Lactobacillales* were below the traditional threshold of 0.05, suggesting potential significance. However, these did not meet the stricter criteria established by the Bonferroni correction for multiple testing, indicating that the observed differences were not statistically significant. Likewise, several commensal genera that were rarely identified or present in small numbers (such as *Corynebacterium*, *Paenarthrobacter*, *Microbacterium*, *Micrococcus*, *Brevibacterium*, *Nocardiopsis*, and *Rhodococcus*) exhibited increased presence after the cessation of disease symptoms. Despite these trends, none of these changes were statistically significant after the correction for multiple comparisons. In contrast, the number of bacteria from the genera *Hafnia*, *Escherichia*, and *Raoultella* remained relatively constant throughout the disease course.

### 3.4. Relationship between the Detection Rates of Opportunistic Bacteria and Demographic and Clinical Factors

Table 3 presents the detection rates of opportunistic bacteria in COVID-19 patients at two different time points: at the onset of the disease (stool sample 1) and 30 days after the cessation of disease symptoms (stool sample 2). The data are stratified by various demographic and clinical characteristics. A statistically significant difference was found in the percentage of opportunistic bacteria between paired samples, with the first stool samples showing a higher mean percentage (mean = 60.26%, SD = 27.10%) compared to the second stool samples (mean = 49.20%, SD = 27.94%) in patients with COVID-19 (*p* = 0.048).

A statistically significant difference was also observed between paired samples from individuals aged 50 to 59 years (*p* = 0.0357), as well as those who did not excrete SARS-CoV-2 through stool (*p* = 0.0332). Although males showed a decrease from a mean percentage of 49.68% to 36.13%, the difference found was not statistically significant (*p* = 0.062). The reduction in opportunistic taxa was not associated with comorbidities, body mass index (BMI), or symptom severity.

Regarding the duration of disease symptoms, two (4%) patients reported a duration of 1–5 days, 10 (20%) reported 6–9 days, 32 (64%) reported 10–14 days, and six (12%) reported more than 14 days. Further examination revealed no significant correlation between the percentage of opportunistic bacteria present in the samples and the duration of COVID-19 symptoms (Figure 3). The correlation coefficients were close to zero (r = −0.177 for sample I and r = −0.025 for sample II), and the *p*-values (*p* = 0.219 for sample I and *p* = 0.862 for sample II) indicated that any observed trends were likely due to random variation rather than a real effect.

### 3.5. Analysis of Inflammatory Biomarkers and SARS-CoV-2 IgG Antibody Levels in COVID-19 Patients

Our study further investigated serum levels of CRP, IL-1α, IL-6, and IL-10 as predictive markers for identifying patients at risk of worsening COVID-19. Figure 4 illustrates the levels of various cytokines and inflammatory markers in patients with COVID-19 at two different time points: within 5 days of the first positive test result (≤5 days) and 30–35 days after the cessation of disease symptoms (30–35 days).

CRP levels exhibited significantly higher values within 5 days of the first positive test compared to 30–35 days after the cessation of disease symptoms, indicating an elevated inflammatory response during the acute phase (Wilcoxon signed-rank test, *p* < 0.001). The variability in CRP levels was also more pronounced in this period, reflecting the dynamic inflammatory status early in the disease course. Additionally, the results indicate a significant increase in IgG antibody levels over time, with *p*-values less than 0.001 at both time points. In contrast, the IL-6, IL-1α, and IL-10 levels showed no statistically significant differences between the two time points. The median value for IL-6 was slightly above 5 pg/mL, with most values clustering around that level. The data spread showed some variability but remained relatively concentrated. The IL-1α levels also exhibited slightly higher variability within 5 days of the first positive test compared to 30–35 days after the cessation of disease symptoms, showing a higher peak at the earlier time point. Additionally, the IL-10 levels showed a similar distribution at both time points, with slightly higher levels within 5 days of the first positive test.

## 4. Discussion

The present study included a cohort of 50 SARS-CoV-2 positive patients receiving outpatient care due to their stable health condition. The research sample was collected between July 2022 and May 2023, a period coinciding with the dominance of the SARS-CoV-2 omicron variant [23]. In Serbia, as well as globally, the omicron variant has led to a significant number of cases, although the incidence of severe illness and mortality has remained relatively low compared to previous viral strains.

The clinical manifestations among the study population were predominantly moderate, characterized by symptoms such as fever, sneezing and cough. Previous investigations into different SARS-CoV-2 variants of concern (VOCs) during earlier waves found that 40 to 85% of infected individuals excreted viral RNA in their feces [24]. In our study, approximately two-thirds (66%) of the omicron SARS-CoV-2 positive patients exhibited viral RNA shedding in their stool at the time of diagnosis, while only one patient (2%) continued to shed the virus 30–35 days after the cessation of disease symptoms. These findings confirm a well-documented observation that, despite omicron causing milder respiratory symptoms [25], the mechanisms underlying fecal virus excretion are likely similar to those of previous VOCs.

Although our study cohort encompassed individuals with mild to moderately severe symptoms, our findings elucidate that during the initial five days of omicron-associated COVID-19, there was a proliferation of opportunistic bacteria at the expense of commensal species, consistent with observations from previous waves [15,16,26]. It is noteworthy that geographical, socio-economic, and cultural factors play significant roles in modulating the composition of the microbial community [27]. Therefore, the predominant bacterial species observed in our study may exhibit certain variations compared to investigations conducted in other regions. Based on the Shannon index, we cannot conclude that there was a decrease in diversity in the fecal samples collected from patients at the onset of the disease and 30–35 days after the cessation of disease symptoms. However, it is evident that the relative abundance of certain taxa changed significantly without affecting overall diversity. Shannon’s and Simpson’s indices primarily assess the overall species richness and evenness within a community, which remained stable despite fluctuations in the relative abundance of individual taxa. The significant compositional changes observed indicate a dynamic response of the gut microbiota to the Omicron infection, likely involving specific taxa rather than a broad alteration in community diversity. Therefore, while the overall diversity metrics did not indicate significant differences, the underlying microbial community composition experienced notable alterations during the course of the disease. Among patients from our region, we observed a reduction in beneficial commensal bacteria belonging to the orders *Bacillales* (including genera such as *Bacillus*, *Lisinibacillus*, and *Psychrobacillus*) and *Lactobacillales* (including genera such as *Lactobacillus*, *Leuconostoc*, *Carnobacterium*, and *Limosilactobacillus*). Specifically, there was a substantial enrichment of the gut microbiota with the genus *Enterococcus* (*E. faecalis*, *E. faecium*) and a significant presence of *Proteobacteria* belonging to the order *Enterobacterales*, such as the *Citrobacter* genus (*C. braakii and C. freundii*).

Enrichment of intestinal microbiota with the species *Enterococcus faecalis* could be highlighted as a significant characteristic of our studied cohort. While we cannot assert that high proportions of enterococci are exclusive to COVID-19, it is worth noting that other studies have also reported an unexpectedly high frequency of enterococci in COVID-19-positive patients [28,29,30]. For instance, in some Italian patients with moderate/severe pneumonia, the gut microbiota was almost mono-dominated by *Enterococcus* spp., mostly *E. faecium*, *E. hirae*, *E. faecalis*, and *E. villorum* [31]. Additionally, our findings regarding the significant presence of *Citrobacter freundii* align with previous studies reporting an increase in *Proteobacteria*, particularly *Citrobacter* spp., in COVID-19 patients. This enrichment of *Citrobacter freundii* may contribute to the dysbiosis observed in COVID-19 patients, potentially impacting the immune response and disease progression [32]. Several mechanisms may underlie the increased prevalence of enterococci in COVID-19 patients. First, it is important to recognize that *Enterococcus* spp. are common inhabitants of the human gastrointestinal tract, capable of utilizing a wide range of substrates for growth and adapting to diverse environmental conditions [33]. The SARS-CoV-2 infection can lead to significant alterations in gut metabolism, including changes in nutrient availability and gut pH. Their exceptional adaptability may allow enterococci to gain a competitive advantage over other bacterial taxa under these altered metabolic conditions [34,35]. Moreover, SARS-CoV-2 infection is associated with considerable immune dysregulation, including altered cytokine profiles and impaired local immune responses, which can suppress the growth of competing bacteria and reduce mucosal immunity. This immune imbalance may facilitate the proliferation of *Enterococcus* spp. [36]. Therefore, it is reasonable to hypothesize that SARS-CoV-2-induced intestinal dysbiosis creates a niche that favors the proliferation of enterococci [31,36,37]. The synergy between SARS-CoV-2 and enterococci may have clinical implications. SARS-CoV-2 can infect enterocytes in the gut, potentially compromising the gut barrier and allowing the translocation of gut bacteria into the bloodstream [31,36,37]. Similarly, the presence of *Citrobacter freundii*, another opportunistic pathogen, could exacerbate gut dysbiosis and contribute to systemic inflammation, further complicating the COVID-19 management [32].

Additionally, the genus *Enterococcus* includes some of the most virulent bacterial strains implicated in human infections, such as *Enterococcus faecium* (known for its resistance to vancomycin) and *Enterococcus faecalis* (which shows resistance to linezolid) [36]. This resistance can pose a challenge during COVID-19 management, especially if secondary bacterial infections arise [31]. Furthermore, *Enterococcus* spp. possess various virulence factors that contribute to their pathogenicity, including the ability to form biofilms, produce toxins [33], and resist immune responses. These factors can complicate COVID-19, particularly in hospitalized patients or those with weakened immune systems.

Several studies, including our own, have identified a specific enrichment of opportunistic fungal pathogens following the resolution of COVID-19 symptoms [38]. Our findings indicated that fecal samples collected 30–35 days after the cessation of disease symptoms showed an increased abundance of the genera *Candida*, *Geotrichum*, *Pichia*, and *Mucor*. In contrast, other studies have primarily reported the prevalence of *Mucor* spp. and *Pneumocystis jirovecii*. According to one report, abnormalities in the gut mycobiota composition persisted after recovery in patients with mild and moderate COVID-19 symptoms [39]. There is increasing evidence that one of the major risk factors for invasive fungal infections is the use of corticosteroids. We hypothesize that the use of antibiotics in some participants could have contributed to creating favorable conditions for fungal proliferation.

Our study was conducted during a period when empirical antibiotic treatment for COVID-19 was common, although it is no longer recommended [40]. Despite the initial exclusion criteria, some participants were administered antibiotics after diagnosis and sample collection as part of their routine clinical care, which was beyond our control. It should be noted that the data on the impact of antibiotics on the microbiota of COVID-19 patients are conflicting. Several studies have reported that patients treated with antibiotics showed greater depletion of beneficial bacteria than patients who did not receive antibiotics [13,41]. Conversely, Vestad et al. (2022) found similar microbiota in antibiotic-treated and untreated COVID-19 patients [41]. Comparing our results with those of other studies is challenging due to variability in clinical conditions, disease duration, use of therapeutic drugs, treatment protocols, and general lifestyle and nutrition. In our study, the overall composition of the intestinal microbiota isolated in culture was more diverse 30–35 days after the cessation of disease symptoms compared to the initial phase of the disease, with a slightly higher proportion of commensal genera such as *Bacillus* and *Lactobacillus.* However, fungal pathogens persisted even after SARS-CoV-2 was cleared from the stool. The observed enrichment of opportunistic bacteria, such as *Enterococcus*, and the persistence of fungal pathogens post-recovery highlight the need for ongoing monitoring of gut microbiota in COVID-19 patients. This aligns with existing scientific evidence suggesting that gut dysbiosis can exacerbate COVID-19 symptoms, emphasizing the potential role of probiotics and dietary interventions as adjunct treatments [42]. Several studies support the beneficial effects of nutritional therapy, including the use of probiotics, prebiotics, and synbiotics combined with a healthy diet, on human health. These interventions may help mitigate SARS-CoV-2-related gut dysbiosis, as evidenced by recent findings [43].

Gut microbiota can both modulate and be modulated by cytokines produced in the gut, thereby directly and indirectly influencing immune responses [44]. Notably, the levels of IL-6 and IL-10 did not show statistically significant changes among the patients in our study despite an imbalance in the composition of intestinal microbiota favoring opportunistic bacteria at the onset of the disease. However, the observed variations in CRP and IL-1α levels during the early phase of infection highlight the dynamic changes in inflammatory responses that occurred during the course of the disease. CRP is a highly sensitive but nonspecific biomarker that indicates inflammation, tissue damage, and infection, and numerous studies have reported elevated CRP levels in patients with COVID-19 [16]. Although CRP is generally well correlated with IL-6, our study did not observe significant changes in IL-6 levels. The mean calculated IL-6 concentration was approximately 5 pg/mL, which is consistent with values reported by other researchers for patients with mild to moderate COVID-19 symptoms associated with the omicron variant [45]. SARS-CoV-2 infection of enterocytes may lead to subtle disturbances in the intestinal environment, potentially explaining the localized intestinal response observed in our study [46]. Additionally, factors such as effective immune regulation preventing systemic inflammation, antibiotic use, and individual predispositions likely contribute to maintaining a mild systemic immune response while allowing minor changes in gut microbiota. Moreover, our findings showed that approximately 94% of participants had detectable SARS-CoV-2 IgG antibodies at the beginning of the infection with varying levels of antibody titers. All participants showed a presence of IgG antibodies 30–35 days after symptom resolution. This widespread seroconversion indicates a sufficiently strong immune response, which may have contributed to the observed normalization of gut microbiota over time.

The results of our study highlight several important trends in the detection rates of opportunistic taxa in patients with COVID-19 over time. Overall, a reduction in opportunistic taxa was observed from the initial stage of the disease to 30–35 days after the cessation of disease symptoms across the entire population. This reduction might suggest a potential recovery or normalization of the gut microbiota as the symptoms of the disease subside. Furthermore, a decrease in opportunistic taxa over time was noted in patients taking vitamin D supplements. The significant impact of vitamin D supplementation could indicate its role in restoring a healthy microbiota. Recent studies have confirmed the anti-biofilm activity of vitamin D against tested Gram-negative strains [47]. However, this reduction in the number of opportunistic pathogens did not reach statistical significance in individuals with comorbidities. Chronic diseases often involve a deterioration of one or more physiological functions, which can modulate the course of COVID-19 [48].

This study has several limitations. We utilized MALDI-TOF MS for bacterial species identification, a method that relies on aligning the mass spectrum of the measured isolate with an established database. The absence of a comparative analysis between MALDI-TOF MS results and 16S rRNA gene sequencing represents a notable limitation. It should be noted that the 16S rRNA sequencing method, commonly used in gut microbiota studies, may not be sensitive enough to detect small microbiota alterations [49]. Therefore, future studies should be based on newer techniques such as shotgun metagenome sequencing. Anaerobic bacteria are more difficult to culture, so bacterial species that were identified by using MALDI-TOF MS represent only a part of the community composition of the gut microbiota. Furthermore, not all laboratories have sequencing capabilities due to the relatively high cost of the technology and the need for specialized bioinformatic tools for analysis. Therefore, in the absence of sequencing resources, culture and MALDI-TOF MS identification can provide valuable insight into the gut microbiota. We believe that this limitation did not substantially impact our findings due to MALDI-TOF MS’s demonstrated ability to accurately identify closely related species. Secondly, this study did not include a control group of uninfected individuals due to a high variability in their baseline characteristics, like lifestyle factors and health conditions (i.e., nutrition, comorbidities, immune responses, medications, etc.), which all might have an impact on the composition of microbiota isolated in culture in both health and disease, as demonstrated in previous research [50]. Instead, we used a longitudinal approach collecting samples from all participants at the time of disease onset and during convalescence. Third, a relatively small number of participants may limit the generalizability of our findings to the broader population from which the sample was taken. Future studies with larger cohorts are needed to validate these results. Furthermore, in future research, a combination of traditional diversity indices with principal component analysis (PCA) could reveal new patterns and trends in the microbial community structure, thereby providing a more comprehensive understanding of the microbiota dynamics.

## 5. Conclusions

In conclusion, this study provides valuable insights into the gut microbiota alterations, inflammatory responses, and fecal shedding patterns in patients infected with the omicron SARS-CoV-2 variant. To the best of our knowledge, this is the first study of its kind conducted in Serbia and the region. Despite the milder respiratory symptoms associated with omicron, we found that approximately two-thirds of patients exhibited viral RNA shedding in their stool at the time of diagnosis. This indicates that the mechanisms of fecal virus excretion remain consistent across different SARS-CoV-2 variants of concern (VOCs).

Our findings contribute to the understanding of the gut microbiota in COVID-19 patients by focusing on the identifiable species within the gut, as determined by culture and MALDI-TOF MS identification. It is crucial to highlight the importance of integrating different methodologies to achieve a comprehensive understanding of gut microbial communities. Access to cultured human microbiota offers detailed functional characterization of bacteria and facilitates the discovery of their biological activities during host–bacterial and inter-bacterial interactions in both health and disease.

Additionally, our findings highlight a significant shift in the gut microbiota composition isolated in culture, characterized by a proliferation of opportunistic bacteria such as *Enterococcus* spp. and *Citrobacter* spp. at the expense of beneficial commensal bacteria from the genera *Bacillus* and *Lactobacillus*. Moreover, the enrichment of fungal pathogens after the cessation of disease symptoms might suggest a prolonged microbiota disruption. The reduction in opportunistic bacterial taxa and stabilization of commensal species over time indicates a potential restoration of gut microbial balance as patients recover from the acute phase of the disease. A significant decrease in CRP levels further supports this recovery trend.

This research contributes to the growing body of evidence on the systemic effects of SARS-CoV-2, highlighting the importance of the gastrointestinal tract in COVID-19. Furthermore, these findings emphasize the need for targeted therapeutic strategies to restore a healthy gut microbiota in COVID-19 patients, potentially involving probiotics, prebiotics, and dietary interventions. Likewise, we believe that this study raises several important questions and provides a basis for conducting research on a larger cohort of patients in our region in order to fully understand the long-term effects of COVID-19 on gut health and the overall health of patients.

## Figures and Tables

**Figure 1 microorganisms-12-01800-f001:**
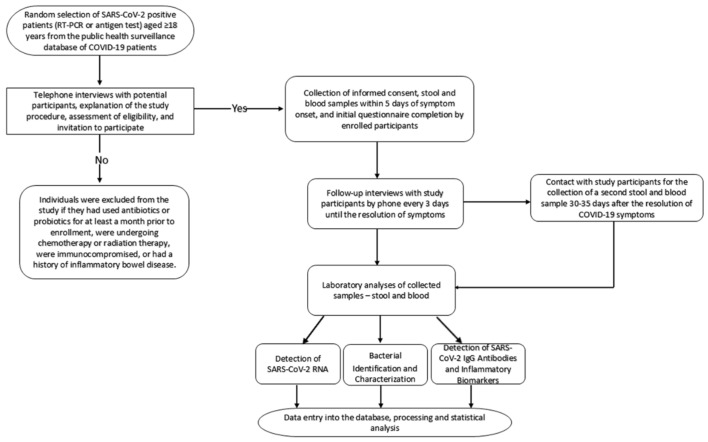
Overview of the study procedure and general sampling process.

**Figure 2 microorganisms-12-01800-f002:**
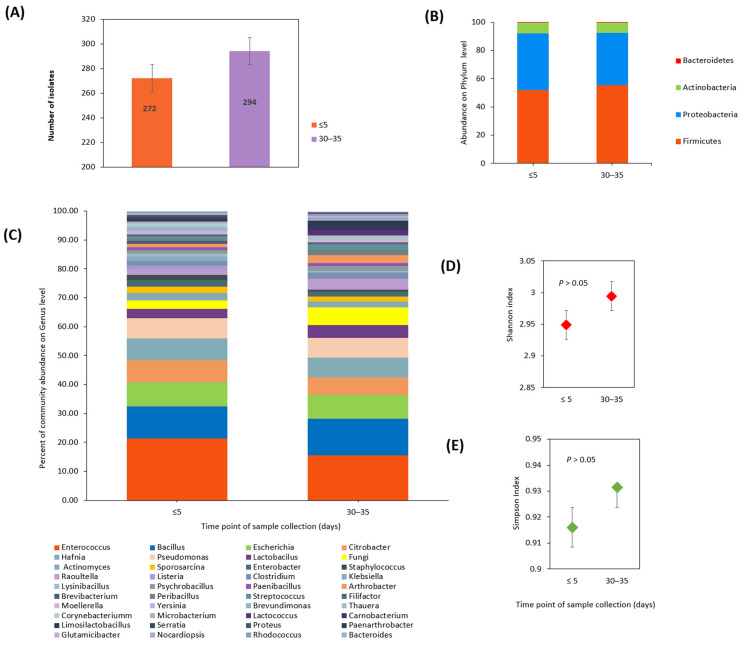
Fecal microbial composition isolated in culture and diversity in samples from SARS-CoV-2 positive patients. Fecal samples were taken at two different time points: within 5 days of the first positive test result for COVID-19 (marked as ≤5 days) and 30–35 days after the cessation of disease symptoms (marked as 30–35 days). (**A**): A total of 272 microbial isolates were identified at the onset of infection (≤5 days), while 294 isolates were identified 30–35 days after the cessation of disease symptoms. Stacked bar plots summarize the relative abundance of microbial taxa at the phylum level (**B**) and genus level (**C**) in samples taken at the two time points. Gut microbiota diversity was assessed using Shannon’s index (**D**) and Simpson’s index (**E**). Both indices showed no statistically significant differences in diversity between the two groups (*p* > 0.05; *t*-test).

**Figure 3 microorganisms-12-01800-f003:**
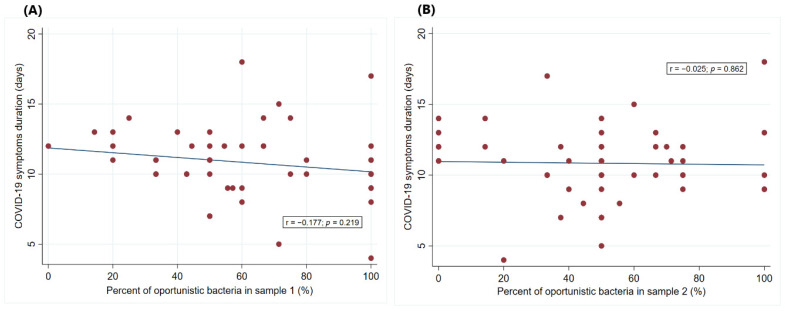
Correlation between the percentage of opportunistic bacteria and the COVID-19 symptom duration. The percentage of opportunistic bacteria in stool samples from COVID-19 patients was measured at two different time points: (**A**) at the onset of the disease (Sample 1) and (**B**) 30–35 days after the cessation of disease symptoms (Sample 2).

**Figure 4 microorganisms-12-01800-f004:**
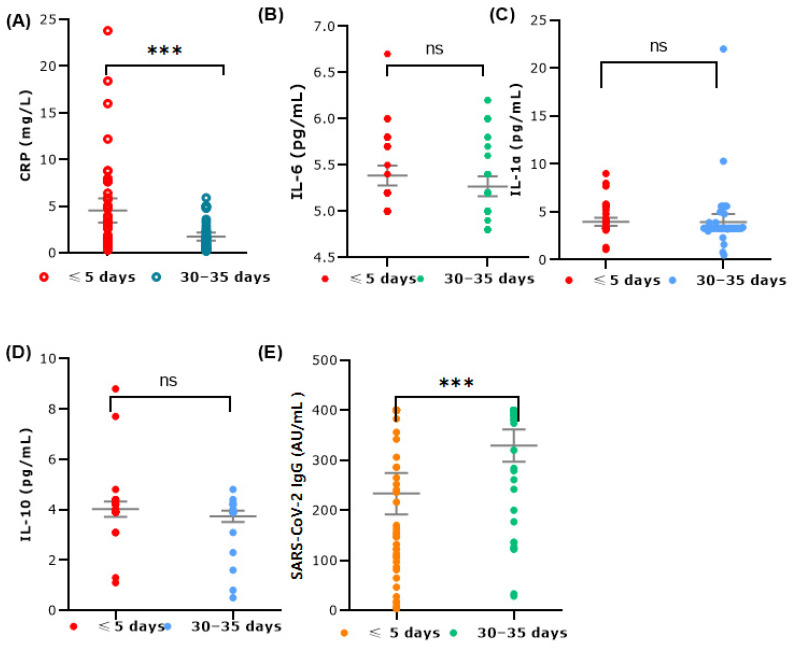
Levels of inflammatory biomarkers and SARS-CoV-2 IgG antibodies in COVID-19 positive patients. The levels of various biomarkers were measured in serum samples at two different time points: ≤5 days and 30–35 days. (**A**) The bar plot shows CRP levels; (**B**) The dot plot compares IL-6 levels; (**C**) The dot plot for IL-1α; (**D**) The dot plot for IL-10 levels. (**E**) The dot plot for SARS-CoV-2 IgG. The plots include individual data points, mean values (central lines), and variability indicators (mean 95% CI). Statistically significant differences between the two groups were observed for the levels of CRP and SARS-CoV-2 IgG according to the Wilcoxon signed-rank test, *** *p* < 0.001; ns: not significant.

**Table 1 microorganisms-12-01800-t001:** Demographic and clinical characteristics of the study cohort in relation to SARS-CoV-2 fecal shedding in samples collected at the onset of infection.

Parameters	N	SARS-CoV-2 Fecal Shedding (*n* = 33; 100%) ^1^	Without SARS-CoV-2 Fecal Shedding (*n* = 17; 100%) ^1^	*p*-Value
**Gender**				
Male	22	12 (36.4%)	10 (58.8%)	0.13
Female	28	21 (63.6%)	7 (41.2%)	
**Age group**				
18–29	2	2 (6.1%)	0	0.386
30–39	9	5 (15.2%)	4 (23.5%)	
40–49	12	10 (30.3%)	2 (11.8%)	
50–59	15	8 (24.2%)	7 (41.2%)	
60+	12	8 (24.2%)	4 (23.5%)	
**COVID-19 severity**				
Asymptomatic	2	1 (3.0%)	1 (5.9%)	0.684
Mild illness	32	23 (69.7%)	9 (52.9%)	
Moderate illness	12	7 (21.2%)	5 (29.4%)	
Severe illness	4	2 (6.1%)	2 (11.8%)	
**COVID-19 duration of symptoms (days)**				
1–5	2	2 (6.1%)	0	0.538
6–9	10	6 (18.2%)	4 (23.5%)	
10–13	32	20 (60.6%)	12 (70.6%)	
≥14	6	5 (15.2%)	1 (5.9%)	
**GI symptoms**				
yes	25	17 (51.5%)	8 (47.1%)	0.765
no	25	16 (48.5%)	9 (52.9%)	
**COVID-19 vaccine status (at least two doses received prior to infection)**				
yes	42	27 (81.8%)	15 (88.2%)	0.558
no	8	6 (18.2%)	2 (11.8%)	
**Prior COVID-19 infection**				
yes	15	8 (24.2%)	7 (41.2%)	0.216
no	35	25 (75.8%)	10 (58.8%)	

^1^ Statistical analysis was performed on samples collected at the onset of infection (≤5 days), given that only one patient continued to shed SARS-CoV-2 in the sample collected 30–35 days after the cessation of disease symptoms.

**Table 2 microorganisms-12-01800-t002:** Statistical analysis overview of the presence and absence of various bacterial and fungal taxa over time.

Bacterial and Fungal Taxa Order/Genus	Bacterial Taxa (Presence or Absence)				Mc Nemar Chi-Squared	*p*-Value
	5 Days/30–35 Days					
	+/+ *n* (%)	+/− *n* (%)	−/+ *n* (%)	−/− *n* (%)		
Genus: *Bacillus* ^1^, *Sporosarcina*, *Lysinibacillus*, *Psychrobacillus*, *Listeria*, *Paenibacillus*, *Peribacillus*	8 (16%)	17 (34%)	6 (12%)	19 (38%)	4.348	0.037
Genus: *Lactobacillus*, *Carnobacterium*, *Filifactor Limosilactobacillus*, *Streptococcus*, *Ligilactobacillus*, *Tissierella*	6 (12%)	5 (10%)	15 (30%)	24 (48%)	4.050	0.044
Genus: *Enterococcus*	31 (62%)	11 (22%)	7 (14%)	1 (2%)	0.500	0.479
Genus: *Pseudomonas*	2 (2%)	7 (14%)	2 (4%)	39 (78%)	1.778	0.182
Genus: *Klebsiella*, *Citrobacter*, *Enterobacter*, *Proteus Moellerella*, *Yersinia*, *Serratia*	8 (14%)	9 (18%)	5 (10%)	28 (56%)	0.643	0.422
Genus: *Escherichia*	14 (26%)	3 (6%)	3 (4%)	30 (60%)	0.167	0.683
Genus: *Hafnia*	8 (16%)	5 (10%)	10 (20%)	27 (54%)	1.067	0.302
Genus: *Raoultella*	6 (12%)	2 (4%)	0 (0.0%)	42 (84%)	1.125	0.289
Genus: *Corynebacterium*, *Micrococcus*, *Paenarthrobacter*, *Microbacterium*, *Brevibacterium*	6 (12%)	4 (8%)	1 ( 2%)	39 (78%)	0.100	0.752
Fungi: *Candida*, *Geotrichum*, *Pichia*, *Mucor*, *Rhodotorula*, *Aspergillus*, *Cryptococcus*	4 (8%)	2 (4%)	10 (20%)	34 (68%)	4.083	0.043

^1^ The genus *Bacillus* was analyzed excluding *Bacillus cereus*, which is considered an opportunistic pathogen. The table analyzes four categories indicating the presence or absence of bacterial and fungal taxa at two different time points: at the first measurement (within 5 days of onset) and at the second measurement (30–35 days after the cessation of disease symptoms). A mark of “+/+” indicates the taxon was observed in both samples; “+/−” indicates the taxon was found only in the first sample; “−/+” indicates the taxon was found only in the second sample; “−/−” indicates the taxon was not found in either sample. Results are presented along with McNemar’s chi-square test and Bonferroni corrected significance, considering a *p*-value below 0.005 as statistically significant

**Table 3 microorganisms-12-01800-t003:** Statistical analysis of the detection rates of opportunistic bacteria stratified by demographic and clinical characteristics.

Parameters	N (%)	Detection Rate of Opportunistic Taxa (%); ≤5 Days		Detection Rate of Opportunistic Taxa (%); 30–35 Day		*p*-Value
		Mean %	SD	Mean %	SD	
**Total**	50	60.26	27.10	49.20	27.94	**0.048 ***
**Gender**						
Male	22 (44%)	62.67	19.39	48.62	28.34	0.062
Female	28 (56%)	58.36	32.13	49.66	28.13	0.292
**Age group**						
18–29	2 (4%)	26.67	9.43	54.17	29.46	0.500
30–39	9 (18%)	47.14	29.28	49.60	26.86	0.865
40–49	12 (24%)	53.10	20.27	49.60	33.95	0.752
50–59	15 (30%)	69.92	24.98	51.11	19.77	**0.036 ***
60+	12 (24%)	70.77	28.03	45.28	34.60	0.063
**GI symptoms** ^a^						
Yes	25 (50%)	57.71	28.62	50.67	25.17	0.393
No	25 (50%)	62.80	25.82	47.73	30.91	0.054
**COVID-19 severity**						
Asymptomatic	2 (4%)	65.00	21.21	25.00	35.36	0.500
Mild illness	32 (64%)	55.86	25.66	49.42	29.46	0.364
Moderate illness	12 (24%)	75.51	24.54	56.19	25.27	0.094
Severe illness	4 (8%)	47.32	38.17	38.57	16.86	0.652
**COVID-19-duration of symptoms (days)**						
1–5	2 (4%)	85.71	20.20	35.00	21.21	0.333
6–9	10 (20%)	68.27	22.22	60.25	23.32	0.378
10–14	32 (64%)	55.02	28.46	47.81	28.18	0.328
≥14	6 (12%)	66.35	24.43	42.94	35.65	0.213
**Comorbidities ^b^**						
Yes	25 (50%)	60.86	28.23	48.59	30.12	0.158
No	25 (50%)	59.65	26.49	49.81	26.18	0.182
**COVID-19 vaccine status (at least two doses)**						
Yes	42 (84%)	60.63	26.39	48.92	28.23	0.056
No	8 (16%)	58.28	32.51	50.69	28.13	0.621
**Prior COVID-19 Infection**						
Yes	15 (30%)	53.95	26.63	47.17	24.92	0.436
No	35 (70%)	62.96	27.23	50.07	29.44	0.071
**SARS-CoV-2 RNA fecal shedding ^c^**						
Yes	18 (36%)	54.84	30.88	49.07	26.82	0.588
No	32 (64%)	63.30	24.73	49.27	28.97	**0.033 ***
**Vitamin D Supplements**						
Yes	17 (34%)	59.35	27.62	60.78	28.24	0.883
No	33 (66%)	60.73	27.25	43.23	26.24	**0.011 ***

^a^—Gastrointestinal (GI) symptoms included: nausea, vomiting, diarrhea; ^b^—Comorbidities (hypertension, diabetes, chronic lung disease, chronic heart disease, chronic kidney disease, or autoimmune disease); ^c^—SARS-CoV-2 RNA fecal shedding observed in the sample collected within the initial 5 days of the onset of COVID-19. Statistical significance was assessed by * paired *t*-test.

## Data Availability

The original contributions presented in the study are included in the article/Appendix A, further inquiries can be directed to the corresponding author.

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
