# Peer review of "Analysis of Cultured Gut Microbiota Using MALDI-TOF MS in COVID-19 Patients from Serbia during the Predominance of the SARS-CoV-2 Omicron Variant"

_microorganisms, 2024, doi:10.3390/microorganisms12091800_

Round 1

Reviewer 1 Report (Previous Reviewer 1)

Comments and Suggestions for Authors

REVIEW

Dear authors,

The paper presents an overview of the impact suffered during the period of predominance of the Omicron variant of SARS-CoV-2 in Serbia, specifically at the gastrointestinal level using mass spectrometry technology for the identification of cultivable microorganisms from the feces of infected patients, however, I consider that corrections should be made in the writing of the paper.

Please amend the requested comments and submit the revision file.

1.    The title should be supplemented with the term “isolated in culture” since mass spectrometry technology is not sufficiently capable of identifying the entire intestinal microbiota, so simply mentioning “Analysis of Gut Microbiota” is incorrect. It should be supplemented with “isolated in culture” so that it is consistent with what was evaluated and presented as results in the work.

2.    Lines 18, 30, 78, 82, 248, 466, 483, 519, 554, 57: should be supplemented with the word “isolated in culture” when mentioning “gut microbiota composition”, in section 4. Discussion they mention the limitations of the study indicating that it was not possible to isolate and identify anaerobic microorganisms, so they limited themselves only to aerobic microorganisms, which they completely omit when mentioning only the term “gut microbiota”.

3.    They do not present the results of the determination of anti-SARS-CoV-2 IgG antibodies that are described in section 2.4 of the results. This result is very important to correlate the increase in antibodies with the change in the microbiota; they must present these results.

4.    Line 306: write the name in cursive “Bacillus”.

5.    Line 574: write the name correctly “Lactobacillus”.

6.    Figure 3: improve the quality of the images, they look pixelated.

Please amend the requested comments and submit the revision file.

Author Response

Response to Reviewer 1

Dear authors,

The paper presents an overview of the impact suffered during the period of predominance of the Omicron variant of SARS-CoV-2 in Serbia, specifically at the gastrointestinal level using mass spectrometry technology for the identification of cultivable microorganisms from the feces of infected patients, however, I consider that corrections should be made in the writing of the paper.

Please amend the requested comments and submit the revision file.

Thank you very much for your careful reading of our manuscript and for your constructive suggestions. We appreciate your detailed feedback, which was crucial in improving the quality of our work. We have revised our manuscript according to your comments and suggestions. Below, we provide a point-by-point response detailing the revisions made.

1. The title should be supplemented with the term “isolated in culture” since mass spectrometry technology is not sufficiently capable of identifying the entire intestinal microbiota, so simply mentioning “Analysis of Gut Microbiota” is incorrect. It should be supplemented with “isolated in culture” so that it is consistent with what was evaluated and presented as results in the work.

Response 1: Thank you for your constructive suggestion regarding the title and for pointing out the need for clarity regarding the methodology used in our study. We revised the title to accurately reflect the scope of our analysis.

The new title is: Analysis of Cultured Gut Microbiota using MALDI TOF MS in COVID-19 Patients from Serbia during the Predominance of the SARS-CoV-2 Omicron Variant

We believe that this modification provides a clearer understanding of our work and aligns with the results presented in the manuscript.

2. Lines 18, 30, 78, 82, 248, 466, 483, 519, 554, 57: should be supplemented with the word “isolated in culture” when mentioning “gut microbiota composition”, in section 4. Discussion they mention the limitations of the study indicating that it was not possible to isolate and identify anaerobic microorganisms, so they limited themselves only to aerobic microorganisms, which they completely omit when mentioning only the term “gut microbiota”.

Response 2: We appreciate your attention to detail and agree that including the phrase “isolated in culture” adds precision. We have added the phrase “isolated in culture” throughout the mentioned lines, as you suggested.

3. They do not present the results of the determination of anti-SARS-CoV-2 IgG antibodies that are described in section 2.4 of the results. This result is very important to correlate the increase in antibodies with the change in the microbiota; they must present these results.

Response 3: Thank you for this constructive suggestion. As recommended, we have included a new chart (Figure 3E) showing SARS-CoV-2 IgG antibody levels (AU/mL) at two different time points: within 5 days of diagnosis and 30-35 days post-diagnosis. The results indicate a significant increase in IgG antibody levels over time, with p-values less than 0.001 at both intervals, confirming the statistical significance of our findings. These data have been incorporated into section 3.5, which has been retitled: "3.5. Analysis of Inflammatory Biomarkers and SARS-CoV-2 IgG Antibodies Levels in COVID-19 Patients" to reflect its broader scope.

Please see the revised section 3.5 (lines 348-383).

 4. Line 306: write the name in cursive “Bacillus”.

Response 4: Thank you for your detailed review and for identifying these typographical errors that we missed in our manuscript. We have made this correction.

5. Line 574: write the name correctly “Lactobacillus”.

Response 5: Thank you for pointing this out. We have corrected the spelling of “Lactobacilus” to “Lactobacillus”.

6. Figure 3: improve the quality of the images, they look pixelated.

Response 6: Thank you for this suggestion. We have improved the quality of Figure 3 to make it clearer.

Please amend the requested comments and submit the revision file.

Once again, we sincerely thank you for all your comments, which have significantly contributed to improving the quality of our manuscript.

Reviewer 2 Report (Previous Reviewer 3)

Comments and Suggestions for Authors

Interesting.

1. Please change "Various factors, such as nutrition habits, genetics, environment and lifestyle ..." to "Various factors, such as nutrition genetics, environment and lifestyle ..."

2. Adding error bars representing standard error or confidence intervals in all figures would improve the clarity of the presented data. This is particularly important in figures displaying microbiota diversity and cytokine levels.

3. The analysis presented in Figure 2 lacks meaning since the variability in symptom duration is likely influenced by numerous factors beyond gut microbiota composition. Instead, the authors can consider dividing the patients into groups based on symptom duration (e.g., <5 days, 6-9 days, 10-13 days, ≥14 days) and compare the relative abundance of opportunistic bacteria between these groups using ANOVA or Kruskal-Wallis tests, depending on data distribution.

4. ". The absence of a comparative analysis between MALDI TOF MS results and 16S rRNA gene sequencing represents a notable limitation." - The 16S rRNA sequencing method, which is used by most gut microbiota studies, may not be sensitive enough to detect small microbiota alterations (citation: pubmed.ncbi.nlm.nih.gov/37110143). This should be mentioned.

Comments on the Quality of English Language

Moderate edits necessary.

Author Response

Response to Reviewer 2

Thank you for your appreciation and for your constructive comments, which have been addressed accordingly. We have thoroughly revised our manuscript based on your comments and suggestions. Below is a point-by-point response detailing the revisions made:

1. Please change "Various factors, such as nutrition habits, genetics, environment and lifestyle ..." to "Various factors, such as nutrition genetics, environment and lifestyle ..."

Response 1: Thank you for this suggestion. We have made the requested change in the manuscript. The sentence now reads: "Various factors, such as nutrition, genetics, environment, and lifestyle ..."

2. Adding error bars representing standard error or confidence intervals in all figures would improve the clarity of the presented data. This is particularly important in figures displaying microbiota diversity and cytokine levels.

Response 2: Thank you for this constructive suggestion. We have revised Figure 3 (in revised manuscript Figure 4) to include error bars representing the 95% confidence intervals, which clarify the variability and precision of the mean cytokine, inflammatory marker, and antibody levels. Appropriate changes have also been made to the text below the chart. Please, see line 368-383.

3. The analysis presented in Figure 2 lacks meaning since the variability in symptom duration is likely influenced by numerous factors beyond gut microbiota composition. Instead, the authors can consider dividing the patients into groups based on symptom duration (e.g., <5 days, 6-9 days, 10-13 days, ≥14 days) and compare the relative abundance of opportunistic bacteria between these groups using ANOVA or Kruskal-Wallis tests, depending on data distribution.

Response 3: Thank you for this suggestion. As it is already presented in the Table 3 where we analyzed in detail the detection rates of opportunistic bacteria stratified by demo-graphic and clinical characteristics, there was 4% of patients with symptoms duration 1-5 days, 20% with 6-9 days, the majority had 10-13 days of symptoms and finally 12% had the duration over two weeks. In this anaylsis we already used the paired t-test (paired samples from each participant in two time points) to compare differences in the relative abundance of opportunistic bacteria between samples collected at the beginning of the disease and 30-35 days after symptoms resolution for each category of symptoms duration, where we found no statistical significance across all explored groups (please see Table 3, lines 314-330). Nevertheless, we followed the reviewer’s suggestion and analyzed the differences between these groups within the first sample and those within the sample two, using the suggested ANOVA method. We noticed that (still) there was no statistically significant difference in the relative abundance of opportunistic bacteria between groups of patients with different duration of symptoms at the onset of disease (p=0.2508) nor at around one month after the symptoms resolution (p=0.4942). We agree with the Reviewer’s comment that variability in symptoms duration is influenced by numerous factors beyond gut microbiota composition and vice versa, but nevertheless we decided to keep the Figure 2 in the manuscript because the number of participants across these groups by duration of symptoms was relatively small to make any meaningful conclusion when making ANOVA analysis, and also because it presents the general direction of the correlation between the percentage of opportunistic bacteria and the COVID-19 symptom duration across two periods of COVID-19, i.e., active part and convalescence.

COVID-19-duration of symptoms (days)

1-5 days

6-9 days

10-13 days

≥14 days

P-value*

N

2 (4%)

10(20%)

32 (64%)

6 (12%)

Sample I

mean

85.71

68.27

55.02

66.35

0.2508

SD

20.2

22.22

28.46

24.43

Sample II

mean

35

60.25

47.81

42.94

0.4942

SD

21.21

23.32

28.18

35.65

*using ANOVA 

4. "The absence of a comparative analysis between MALDI TOF MS results and 16S rRNA gene sequencing represents a notable limitation." - The 16S rRNA sequencing method, which is used by most gut microbiota studies, may not be sensitive enough to detect small microbiota alterations (citation: pubmed.ncbi.nlm.nih.gov/37110143). This should be mentioned.

Response 4: Thank you for your insightful comment.  We have added the following sentence to the limitations section to address this point and included the relevant reference you suggested:

..... results and 16S rRNA gene sequencing represent a notable limitation. "It should be noted that the 16S rRNA sequencing method, commonly used in gut microbiota studies, may not be sensitive enough to detect small microbiota alterations [49]. Therefore, future studies should be based on newer techniques such as shotgun metagenome sequencing." Please, see line 548-552.

Reviewer 3 Report (New Reviewer)

Comments and Suggestions for Authors

A flow chart would be helpful to indicate the sampling of patients and the procedures.

Background data on patients, such as symptoms, comorbidities, treatment during COVID-19, etc., are missing.

It may be difficult, but it would be interesting to have each patient's Omicron variant typing and compare it with the microbiota.

Expand the discussion to include a detailed comparison with previous studies and the implications of the findings for theory and practice.

Discuss the possible limitations of the study and suggest future research directions.

Discuss how the results may influence clinical practices or future research in microbiota and COVID-19.

Comments on the Quality of English Language

Minor editing of English language required

Author Response

Response to Reviewer 3

Thank you for your appreciation and for your constructive comments, which have been addressed accordingly. We have thoroughly revised our manuscript based on your comments and suggestions. Below is a point-by-point response detailing the revisions made:

1. A flow chart would be helpful to indicate the sampling of patients and the procedures.

Response 1: Thank you for this useful suggestion. We prepared the flowchart that depicts the overview of the recruitment process and sampling procedure, please see Figure 1.

2. Background data on patients, such as symptoms, comorbidities, treatment during COVID-19, etc., are missing.

Response 2: Thank you for highlighting this important aspect. We recognize that patient clinical characteristics, symptoms, comorbidities, and treatments can significantly affect study results. To comprehensively address your concern, we have added a brief summary indicating the sections of the manuscript that describe these features. Specifically, we have presented the study cohort data in both textual form and in Table 1. The textual presentation includes detailed information on the demographic and clinical characteristics of the cohort, such as disease severity distribution, vaccination status, and the presence of SARS-CoV-2 RNA in stool samples. Please, see lines 185-212.

Additionally, in section 3.4, "Relationship between the Detection Rates of Opportunistic Bacteria and Demographic and Clinical Factors," we have included comorbidities as one of the variables in the statistical analysis. Explanations for these combinations can be found in the footnote below the table. Please, see lines 312-334.

Moreover, we acknowledge that patient treatment data were not included, as this aspect was managed by the health center physicians and was outside the scope of our study.

3. It may be difficult, but it would be interesting to have each patient's Omicron variant typing and compare it with the microbiota.

Response 3: Thank you for this interesting suggestion. Although the cost of next-generation sequencing has started to decrease in the last few years, our budget was limited during the period when the study was conducted. Determining the circulating Omicron sub-variants and comparing them with the intestinal microbiota would indeed be an interesting avenue of research. We appreciate the idea and may consider it for future research.

4. Expand the discussion to include a detailed comparison with previous studies and the implications of the findings for theory and practice.

Response 4: Thank you for your insightful comments and suggestions. In our revised discussion, we have addressed these aspects as follows:

"Additionally, our findings regarding the significant presence of Citrobacter freundii align with previous studies reporting an increase in Proteobacteria, particularly Citrobacter spp., in COVID-19 patients. This enrichment of Citrobacter freundii may contribute to the dysbiosis observed in COVID-19 patients, potentially impacting the immune response and disease progression [32]. "

Please, see line 434-439

…….. "Similarly, the presence of Citrobacter freundii, another opportunistic pathogen, could exacerbate gut dysbiosis and contribute to systemic inflammation, further complicating the COVID-19 management [32]. "

Please, see line 454-457

"The observed enrichment of opportunistic bacteria, such as Enterococcus, and the persistence of fungal pathogens post-recovery highlight the need for ongoing monitoring of gut microbiota in COVID-19 patients. This aligns with existing scientific evidence suggesting that gut dysbiosis can exacerbate COVID-19 symptoms, emphasizing the potential role of probiotics and dietary interventions as adjunct treatments [41] Several studies support the beneficial effects of nutritional therapy, including the use of probiotics, prebiotics, and synbiotics combined with a healthy diet, on human health. These interventions may help mitigate SARS-CoV-2-related gut dysbiosis, as evidenced by recent findings [42]."

Please, see line 495-503

We have introduced three new references to support these points and provide a comprehensive discussion by comparing our findings with previous studies and outlining the implications for both theory and practice. We believe that our current revisions sufficiently address and align with your suggestion.

5. Discuss the possible limitations of the study and suggest future research directions.

Response 5: Thank you for your insightful comment. We have discussed the limitations of our study and outlined future research directions in the manuscript. Specifically, we have acknowledged that using MALDI-TOF MS for bacterial species identification has limitations compared to 16S rRNA gene sequencing. Additionally, we have mentioned the relatively small sample size, which may limit the generalizability of our findings. For future research, we have suggested larger cohort studies, newer techniques such as shotgun metagenome sequencing, and the combination of traditional diversity indices with Principal Component Analysis (PCA) to provide a more comprehensive understanding of microbiota dynamics. These points are detailed in the "Study Limitations" section.

6. Discuss how the results may influence clinical practices or future research in microbiota and COVID-19.

Response 6: Thank you for this suggestion. We have discussed the potential impact of our findings on clinical practice and future research in the "Discussion" (please, see response 4) and "Conclusions" section as follows:

"Furthermore, these findings emphasize the need for targeted therapeutic strategies to restore a healthy gut microbiota in COVID-19 patients, potentially involving probiotics, prebiotics, and dietary interventions."

Please, see line 602-605

We have also emphasized the importance of integrating different methodologies to achieve a comprehensive understanding of gut microbial communities. Additionally, our findings provide a basis for conducting larger cohort studies to understand the long-term effects of COVID-19 on gut health.

Round 2

Reviewer 1 Report (Previous Reviewer 1)

Comments and Suggestions for Authors

Dears authors,

The suggested modifications and corrections were made to improve the quality of the work. By adding the term "isolated in culture" it is understood that it is not a conventional microbiota analysis by sequencing, but rather comes from the traditional culture of the isolated microorganisms.

This manuscript is a resubmission of an earlier submission. The following is a list of the peer review reports and author responses from that submission.

Round 1

Reviewer 1 Report

Comments and Suggestions for Authors

REVIEW

Dear authors,

The work shows the changes in the intestinal microbiota of patients infected with SARS-CoV-2 during the period of predominance of the Omicron variant in Serbia, as well as its correlation with clinical and immunological aspects. The present work proposes the use of mass spectrometry technology for the identification of microorganisms isolated from cultures; however, I consider that there are several methodological questions that must be justified.

Please amend the requested comments and submit the revision file.

1.    The title has a great deficiency with respect to what is presented in the work, since it mentions “gut microbiota” and what is actually being evaluated in patients is only the culturable microbiota, leaving aside everything that is not cultivable. On the other hand, the study has a very small cohort (50 patients) and they are specifically from Serbia, so if the origin of the patients is not defined in the title it can be confused with the fact that the work was carried out with a much larger cohort and that encompasses multiple regions of the world. They should correct the title.

2.    Although they mention it in the conclusions, the methodology used MALDI-TOF MS does not have the same capacity as 16S rRNA sequencing, so I do not consider that the scope of the work can reach an “analysis of the intestinal microbiota”, it is should focus only on the “identification of commensal bacteria and fungi with pathobiont potential” when conditions are suitable.

3.    The methodology for isolating microorganisms from feces is not well described. They mention that a portion of the sample is processed to obtain a 10% suspension, but how much does that portion of the sample weigh? Is it the same weight for everyone? How did they define it? Because they were able to make an initial dilution and make dilutions to subsequently sow aliquots on the plates, the results will be quantifiable and in this way they could find differences in the abundances of culturable microorganisms and subsequently identify them by MALDI-TOF MS.

4.    Was the result of the molecular detection of SARS-CoV-2 by the PCR kit quantifiable? This would help correlate the viral load present in the feces with the change in the isolated microorganisms.

5.    Did you only use one type of culture medium to isolate the microorganisms from the samples?

6.    If the total number of patients is small (n=50) it cannot be considered representative that there were only 2 patients in the age range of 18-29 years. This represents a great deficiency in the work.

7.    The way in which they obtained the percentages of relative abundances at the phylum and genus level is confusing since how they represent it is typical of results obtained from sequencing, when in reality they were not obtained by this technique.

8.    When mentioning that the sampling was carried out during the circulation of the Omicron variant of concern, they had to corroborate by sequencing that it corresponded to this variant.

9.    There is no supplementary material (Table 1S).

10. Write the names of the microorganisms correctly since several are not written in cursive.

Please amend the requested comments and submit the revision file.

Author Response

Response to Reviewer 1

Dear authors,

The work shows the changes in the intestinal microbiota of patients infected with SARS-CoV-2 during the period of predominance of the Omicron variant in Serbia, as well as its correlation with clinical and immunological aspects. The present work proposes the use of mass spectrometry technology for the identification of microorganisms isolated from cultures; however, I consider that there are several methodological questions that must be justified.

Please amend the requested comments and submit the revision file.

Response: We thank the Reviewer for the appreciation and for constructive comments, which have been addressed accordingly. We have thoroughly revised our manuscript in accordance with your comments and suggestions. Below, we provide a point-by-point response detailing the revisions made:

1. The title has a great deficiency with respect to what is presented in the work, since it mentions “gut microbiota” and what is actually being evaluated in patients is only the culturable microbiota, leaving aside everything that is not cultivable. On the other hand, the study has a very small cohort (50 patients) and they are specifically from Serbia, so if the origin of the patients is not defined in the title it can be confused with the fact that the work was carried out with a much larger cohort and that encompasses multiple regions of the world. They should correct the title.

Response 1: Thank you for this comment. We have considered your suggestion to make the title more reflective of the focus on culturable microorganisms and the specific cohort studied. In response to your comment, we have revised the title to:

Analysis of Gut Microbiota using MALDI TOF MS in COVID-19 Patients from Serbia during the Predominance of the SARS-CoV-2 Omicron Variant

2. Although they mention it in the conclusions, the methodology used MALDI-TOF MS does not have the same capacity as 16S rRNA sequencing, so I do not consider that the scope of the work can reach an “analysis of the intestinal microbiota”, it is should focus only on the “identification of commensal bacteria and fungi with pathobiont potential” when conditions are suitable.

Response 2:  Thank you for your feedback. We appreciate your point regarding the comparative capabilities of MALDI-TOF MS and 16S rRNA sequencing. Our study employed MALDI-TOF MS, which focuses on identifying bacterial and fungal species based on protein profiling. We used the term "gut microbiota", which describes the broader ecological context referring to the microbial community residing in the gut. To address your concern and avoid overstating our results, we have clarified that our study focuses on the "identifiable portion of the gut microbiota using MALDI-TOF MS". We hope that this adjustment aligns with your expectations and provides a clearer representation of our research.

Consequently, we have revised the Abstract (line 23-24), Introduction (line 81-84), and Conclusion sections (line 574-584) to more clearly emphasize that the identified microorganisms are those detectable by MALDI-TOF MS.

3. The methodology for isolating microorganisms from feces is not well described. They mention that a portion of the sample is processed to obtain a 10% suspension, but how much does that portion of the sample weigh? Is it the same weight for everyone? How did they define it? Because they were able to make an initial dilution and make dilutions to subsequently sow aliquots on the plates, the results will be quantifiable and in this way they could find differences in the abundances of culturable microorganisms and subsequently identify them by MALDI-TOF MS.

Response 3: Thank you for highlighting this aspect, allowing us to improve the transparency of our methodology. We agree that serial dilutions could provide a quantifiable data on microbial abundance. However, due to limited resources, particularly the larger quantities of culture media required for serial dilutions, our study focused only to a qualitative analysis of the microbial composition. This approach was selected to ensure consistency within the constraints of our available resources.

We have revised the Methods section of our manuscript to include these details for clarity, as follows:

"Immediately upon arrival at the laboratory, fecal samples were processed to create a 10% suspension by homogenizing 1 gram of fecal material in 9 mL of sterile balanced salt solution (0.9% NaCl) while the remainder was sent for microbiological analysis". Please, see line 113-116;  and  line 124-138.

4. Was the result of the molecular detection of SARS-CoV-2 by the PCR kit quantifiable? This would help correlate the viral load present in the feces with the change in the isolated microorganisms.

Response 4: We used a qualitative PCR kit that detects the presence of SARS-CoV-2 RNA in fecal samples without providing viral load quantification.

5. Did you only use one type of culture medium to isolate the microorganisms from the samples?

Response 5Thank you for the comment. Stool specimens were inoculated onto several nutrient media, including Columbia Blood agar, Endo agar, Salmonella Shigella (SS) Agar, Schaedler agar, and Sabouraud agar.

Following your comment, we have added additional information, please see page 3, line 134-136.

6. If the total number of patients is small (n=50) it cannot be considered representative that there were only 2 patients in the age range of 18-29 years. This represents a great deficiency in the work.

Response 6: Thank you for your feedback regarding the age distribution in our study sample. We acknowledge that the small sample size (n=50), particularly the limited representation of patients in the 18-29 age range (only 2 patients), restricts the generalizability of our findings. We stated this in the limitation of the study (please, see line 550-552). This age distribution may limit the ability to draw broad conclusions about the impact of COVID-19 across different age groups.

We believe that acknowledging this limitation provide context for interpreting our results and highlight the importance of further research to confirm our observations in a more representative population.

7. The way in which they obtained the percentages of relative abundances at the phylum and genus level is confusing since how they represent it is typical of results obtained from sequencing, when in reality they were not obtained by this technique.

Response 7: Thank you for your observation. We would like to clarify that our representation of relative abundances was derived from the proportion of identified taxa using MALDI-TOF MS, not from sequencing. The format of our results might resemble those typically seen in sequencing studies, but it was achieved by calculating the percentages of bacterial taxa (at both the genus and phylum levels) based on the identified species and then visualizing these proportions with a stacked column chart in Excel. To avoid any confusion and to accurately convey the scope of our analysis, we added sentence in part 3.2.

"These findings were visually represented using percentages in a stacked bar plot". Please, see line 244-245.

8. When mentioning that the sampling was carried out during the circulation of the Omicron variant of concern, they had to corroborate by sequencing that it corresponded to this variant.

Response 8: Thank you for your feedback regarding the verification of the Omicron variant during the study period. We acknowledge the importance of confirming the circulating variant for the accuracy of our findings. Our study utilized data from GISAID, accessed via CoVariants.org, which actively monitors SARS-CoV-2 variants globally. According to data presented by Our World in Data, the predominant SARS-CoV-2 variants circulating in Serbia from June 20, 2022, to the present were sub-variants of Omicron, including BA.1, BA.2, BA.2.12.1, and others within the BA lineage.

Given that our sample collection was conducted within this timeframe, and considering that the Omicron variant accounted for the overwhelming majority of cases in Serbia during this period, we concluded that Omicron was the predominant variant impacting our study participants. This epidemiological data supports our assumption that the Omicron variant was the main variant during our study.

To document this information as comprehensively as possible, we have excluded previously reference No.21 and included a new reference (in revised manuscript No. 23) that accurately reflects this data source:

Our World In Data. SARS-CoV-2 sequences by variant, Serbia, Nov 21, 2022. [Accessed June 26, 2024.]. Available from: https://ourworldindata.org/grapher/covid-variants-bar?country=~SRB

9. There is no supplementary material (Table 1S).

Response 9: Thank you for the comment. We added the supplementary material, including Table 1S. Please see the supplementary section of the manuscript for additional data.

10. Write the names of the microorganisms correctly since several are not written in cursive.

Response 10: Thank you for pointing out this oversight. We will ensure that all microorganism names are properly italicized in the revised manuscript.

Please amend the requested comments and submit the revision file.

Once again, we sincerely appreciate your insightful feedback, which has significantly contributed to improving the quality of our manuscript.

Reviewer 2 Report

Comments and Suggestions for Authors

Patic et al studied the gut microbiota differences in omicron variant of SARS-CoV-2 infected individuals. Several studies about gut microbiota in COVID-19 patients are reported earlier, however, here authors enquired whether lesser virulent omicron has any differences in gut microbiome composition. They tested the presence of SARS-CoV2 RNA in stool and cytokines in serum. More importantly they used initial (within 5 days) and final (30-35 days) samples for assessing the composition of microbiota using MULDI-TOF mass spectrometric methods. While the study is important, the following concerns should be considered.

1.       Section 2.5 should be section 2.1

2.       In table 1, line 192 should be properly aligned.

3.       In table 1 legend, the percentage presented within parenthesis of each row are the percentage calculations from total shedding and non-shedding, but not from total sample. Particularly, because it is confusing when N is given in a column.

4.       although it is mentioned in the text, it is recommended to mention within the table,  the results are from the first set of sample collections (~ 5 days) in  (may be below parameters or somewhere),

5.       Line 201, there is no supplementary information nor Table 1S in the manuscript, thus can not be evaluated.

6.       Line 206-208, what refers to “times”. They should explain it in the text.

7.       In line 227, Figure 1d and 1e are already referred, no need to mention figure 1.

8.       In line 247, the name of opportunistic taxa should be given in parenthesis for easy reading.

9.       In table 2, in the heading, a row should be added for 5days/35days for easy understanding of +/+, +/- ,-/+ and -/-, instead of writing in legend.

10.   In figure 3c, what is p/ml in the y axis?

11.   Line 417,421-435, initially it was mentioned that antibiotics users are excluded (line 92) from the study

12.   Line 443, IL-1alpha did change in figure 3C

13.   What is the reason for increasing Enterococcus in COVID-19 patients? In other words, what changes does SARS-CoV-2 make for increasing Enterococcus? Authors should present a possible reason in discussion.

Comments on the Quality of English Language

Minor mistakes in grammar should be corrected

Author Response

Response to Reviewer 2

Patic et al studied the gut microbiota differences in omicron variant of SARS-CoV-2 infected individuals. Several studies about gut microbiota in COVID-19 patients are reported earlier, however, here authors enquired whether lesser virulent omicron has any differences in gut microbiome composition. They tested the presence of SARS-CoV2 RNA in stool and cytokines in serum. More importantly they used initial (within 5 days) and final (30-35 days) samples for assessing the composition of microbiota using MULDI-TOF mass spectrometric methods. While the study is important, the following concerns should be considered.

Thank you very much for your careful reading of our manuscript and for your constructive suggestions. We appreciate your detailed feedback, which was crucial to improving the quality of our work. We have thoroughly revised our manuscript in accordance with your comments and suggestions. Below, we provide a point-by-point response detailing the revisions made:

1. Section 2.5 should be section 2.1

Response 1: Thank you for the suggestion. We have moved the content from Section 2.5 to Section 2.1. Please see the revised version of the manuscript, line 108-110.

2. In table 1, line 192 should be properly aligned.

Response 2: Thank you for your observation. We have corrected the alignment of Table 1. Please see the revised version of Table 1 in the manuscript.

3. In table 1 legend, the percentage presented within parenthesis of each row are the percentage calculations from total shedding and non-shedding, but not from total sample. Particularly, because it is confusing when N is given in a column.

Response 3: Thank you for your feedback. We have clarified the legend by explicitly detailing the percentage calculations. Please see the revised version of Table 1 in the manuscript.

4. although it is mentioned in the text, it is recommended to mention within the table, the results are from the first set of sample collections (~ 5 days) in (may be below parameters or somewhere),

Response 4: Thank you for the recommendation. We agree that specifying the timing of the sample collections directly within the table will improve clarity. We have updated the title of Table 1 to clearly indicate that the results are derived from the first set of sample collections (~5 days) as follows:

"Table 1.Demographic and clinical characteristics of the study cohort in relation to SARS-CoV-2 fecal shedding in samples collected at the onset of infection". Please, see line 205-206.

5. Line 201, there is no supplementary information nor Table 1S in the manuscript, thus can not be evaluated.

Response 5: Thank you for the comment. We have added the supplementary material, including Table 1S. Please see the supplementary section of the manuscript for additional data.

6. Line 206-208, what refers to “times”. They should explain it in the text.

Response 6: Thank you for the suggestion. We have clarified that "times" refers to the number of distinct samples in which each species was detected. The text has been revised as follows:

"Among the 83 identified bacterial species, 13 species (15.7%) were observed in only one sample, 45 species (54.2%) were detected in 2 to 5 different samples, 14 species (16.9%) were found in 6 to 10 different samples, and 11 species (13.2%) were identified in more than 10 distinct samples".

Please, see line 227-229.

7. In line 227, Figure 1d and 1e are already referred, no need to mention figure 1.

Response 7: Thank you for the comment. We have removed the redundant mention of “Figure 1”.

8. In line 247, the name of opportunistic taxa should be given in parenthesis for easy reading.

Response 8: Thank you for the suggestion. We have included the names of opportunistic taxa in parentheses for clarity. Please, see line 271-272.

9. In table 2, in the heading, a row should be added for 5days/35days for easy understanding of +/+, +/- ,-/+ and -/-, instead of writing in legend.

Response 9: Thank you for the suggestion. We have added a row in Table 2 for the 5 days/35 days categorizations to enhance clarity.

10. In figure 3c, what is p/ml in the y axis?

Response 10: Thank you for pointing this out. We have corrected the text error on the y-axis of Figure 3c to accurately represent the data.

11. Line 417,421-435, initially it was mentioned that antibiotics users are excluded (line 92) from the study

Response 11: Thank you for your observation. While one of the inclusion criteria was that patients did not initially take antibiotics or probiotics, it is important to note that some patients did receive these treatments as part of their routine clinical care after being diagnosed with SARS-CoV-2 and after providing blood and stool samples. This treatment decision was made by the doctors providing clinical care and was beyond our control. We have clarified this in the manuscript as follows:

We hypothesize that the use of antibiotics in some participants could have contributed to creating favorable conditions for the fungal proliferation. Our study was conducted during a period when empirical antibiotic treatment for COVID-19 was common, although it is no longer recommended [37]. ...."Despite the initial exclusion criteria, some participants were administered antibiotics after diagnosis and sample collection as part of their routine clinical care, which was beyond our control".

Please see page 15, lines 486-490. 

12. Line 443, IL-1alpha did change in figure 3C

Response 12: Thank you for your observation. We have updated the manuscript to consistently refer to IL-1 alpha as "IL-1α" throughout the text for clarity and accuracy. Specifically, in relation to Figure 3C, we have revised our description to accurately reflect the observed changes in IL-1α levels between the two time points. The updated figure legend and corresponding sections now correctly present the data, showing that IL-1α levels did not significantly change between the early infection phase (≤ 5 days) and the recovery phase (30-35 days).

Please, see Figure 3, figure legend and revised text in manuscript, line 387-393

13. What is the reason for increasing Enterococcus in COVID-19 patients? In other words, what changes does SARS-CoV-2 make for increasing Enterococcus? Authors should present a possible reason in discussion.

Response 13: Thank you for the suggestion. We have added a discussion on the potential reasons for the increased prevalence of Enterococcus spp. in COVID-19 patients, as follows:

"Several mechanisms may underlie the increased prevalence of enterococci in COVID-19 patients. First, it is important to recognize that Enterococcus spp. are common inhabitants of the human gastrointestinal tract, capable of utilizing a wide range of substrates for growth and adapting to diverse environmental conditions [32]. SARS-CoV-2 infection can lead to significant alterations in gut metabolism, including changes in nutrient availability and gut pH. Their exceptional adaptability may allow enterococci to gain a competitive advantage over other bacterial taxa under these altered metabolic conditions [33,34]. Moreover, SARS-CoV-2 infection is associated with considerable immune dysregulation, including altered cytokine profiles and impaired local immune responses, which can suppress the growth of competing bacteria and reduce mucosal immunity. This immune imbalance may facilitate the proliferation of Enterococcus spp. [35]. Therefore, it is reasonable to hypothesize that SARS-CoV-2-induced intestinal dysbiosis creates a niche that favors the proliferation of enterococci [31, 35, 36]". Please see line 446-459.

Reviewer 3 Report

Comments and Suggestions for Authors

1. In the introduction, it should be mentioned that bacterial and viral infections, including SARS-CoV-2 has been associated with post-infectious irritable bowel syndrome (citation: pubmed.ncbi.nlm.nih.gov/38699957).

2. "Given the global predominance of the Omicron variant as the current sole circulating variant ..." - this information is now outdated. Please update the statement to reflect the current epidemiological situation, acknowledging the presence of multiple new circulating variants.

3. "This prospective cohort study included 50 adult patients ..." - how was the sample size decided?

4. The nature of replication in the experimental design is unclear, and the assessment of uncertainty in the reported measurement is absent or unclear. Information on the number of replicates conducted for each experiment and the variation observed would add credibility to the findings.

5. The authors relied on MALDI-TOF mass spectrometry for identifying bacterial species. While this technique is beneficial for its speed and accuracy in clinical settings, it may not be as sensitive as other methods like 16S rRNA or metagenomic sequencing for detecting subtle changes in the gut microbiota. Further elaboration and justification for the choice of method is necessary.

6. The absence of a control group of healthy individuals makes it difficult to attribute changes in the gut microbiota specifically to COVID-19 as opposed to other potential confounding factors. This is a major limitation and should be mentioned.

7. Given the biological nature of the data, which often do not follow a normal distribution, verifying the assumptions of normality is crucial. Non-normally distributed data should ideally be analyzed using non-parametric tests such as the Wilcoxon signed-rank test. 

8. With multiple tests being conducted, the risk of Type I errors (false positives) increases. The study does not mention any correction for multiple comparisons, such as the Bonferroni correction or False Discovery Rate (FDR) control.

9. Employing more comprehensive diversity metrics and community composition analyses, such as Principle Component Analysis (PCA) should be considered. 

10. The study claims a significant change in gut microbiota composition during the initial phase of Omicron infection but reports no significant changes in overall diversity indices (Simpson’s and Shannon’s). This apparent contradiction needs further discussion and clarification.

11. There are several instances of typographical errors and inconsistent formatting throughout the manuscript that need to be corrected (e.g., inconsistent use of units and missing spaces).

Comments on the Quality of English Language

Moderate edits necessary.

Author Response

Response to Reviewer 3

Thank you very much for your careful reading of our manuscript and for your constructive suggestions. We appreciate your detailed feedback, which was crucial to improving the quality of our work. We have thoroughly revised our manuscript in accordance with your comments and suggestions. Below, we provide a point-by-point response detailing the revisions made:

1. In the introduction, it should be mentioned that bacterial and viral infections, including SARS-CoV-2 has been associated with post-infectious irritable bowel syndrome (citation: pubmed.ncbi.nlm.nih.gov/38699957).

Response 1: Thank you for your comment. Following your suggestion, we have added a sentence to the Introduction (page 2, lines 46-48): "Different bacterial and viral infections, including SARS-CoV-2, have previously been associated with persistent gastrointestinal symptoms and post-infectious irritable bowel syndrome." 

Additionally, we have included a new reference in the manuscript under number [9].

2. "Given the global predominance of the Omicron variant as the current sole circulating variant ..." - this information is now outdated. Please update the statement to reflect the current epidemiological situation, acknowledging the presence of multiple new circulating variants.

Response 2: Thank you for this helpful comment. As advised, we have updated the information in the Introduction, please see lines 72-78:

 "The virus has continued to evolve and accumulate mutations, particularly in the receptor-binding domain of the Spike protein. This has led to the simultaneous rise of multiple Omicron descendants, sharing common mutations that enhance the virus's ability to evade neutralizing antibodies. Omicron subvariants currently circulating globally include JN.1, KP.2, KP.3, and KP.1."

...Given the global predominance of the Omicron variant  "and its subvariants" .........

3."This prospective cohort study included 50 adult patients ..." - how was the sample size decided?

Response 3: The sample size of 50 patients was determined based on the number of patients who agreed to participate in the study and could provide both the first and second stool samples, along with a blood sample. Additionally, logistical constraints and the limited resources available from our funding source were also factors considered in determining the sample size.

4. The nature of replication in the experimental design is unclear, and the assessment of uncertainty in the reported measurement is absent or unclear. Information on the number of replicates conducted for each experiment and the variation observed would add credibility to the findings.

Response 4: Thank you for raising this important point. Due to limited financial resources, we were only able to conduct the experiments once. While we acknowledge that repetition is essential for assessing variability and enhancing the robustness of our results, our funding constraints prevented us from performing additional replicates. However, despite this limitation, we meticulously followed established data collection and analysis protocols, adhered to good laboratory practices, and implemented well-established methodologies with quality control measures to ensure the validity and reliability of our findings.

5. The authors relied on MALDI-TOF mass spectrometry for identifying bacterial species. While this technique is beneficial for its speed and accuracy in clinical settings, it may not be as sensitive as other methods like 16S rRNA or metagenomic sequencing for detecting subtle changes in the gut microbiota. Further elaboration and justification for the choice of method is necessary.

Response 5: We appreciate this comment. We agree that highly sensitive methods like 16S rRNA or metagenomic sequencing are preferred for analyzing the microbiota. Our main reason for using culture and MALDI-TOF MS identification was limited financial resources, but we also believe that this method has some advantages. MALDI-TOF MS is a method requiring minimal training and can identify bacterial strains within minutes. Furthermore, culture methods specifically identify viable populations within the gut microbiota, while most molecular methods do not differentiate between DNA derived from live or dead cells. In addition, culture methods employing selective media enable the growth and detection of less abundant bacteria that might be overlooked due to insufficient sequencing depth. Next-generation sequencing approaches, particularly those based on amplification of the highly conserved 16S rRNA gene, are inherently limited in their ability to detect intra-species variations. Additionally, culture methods enable the identification of fungi, such as Candida and Aspergillus, which cannot be detected using 16S rRNA sequencing. This is the first study on the impact of SARS-CoV-2 on the intestinal microflora in Serbia and the region. We believe that the obtained data are significant and provide a basis for conducting further analyses. In our next study, we plan to expand our research to a larger cohort of patients in our region and to employ highly sensitive methods like 16S rRNA or metagenomic sequencing. We are aware of the disadvantages of our current methods, which is why we have included an explanation of this issue in the limitation section of the study, please see page 16, lines 542-548:

"This study has several limitations. We utilized culture and MALDI-TOF MS for bacterial species identification, a method that relies on aligning the mass spectrum of the measured isolate with an established database. The absence of a comparative analysis between MALDI-TOF MS results and 16S rRNA gene sequencing represents a notable limitation. Anaerobic bacteria are more difficult to culture, so bacterial species that were identified by using MALDI-TOF MS represent only a part of the community composition of the gut microbiota. However, we believe that this limitation did not substantially impact our findings due to MALDI-TOF MS's demonstrated ability to accurately identify closely related species. "

6. The absence of a control group of healthy individuals makes it difficult to attribute changes in the gut microbiota specifically to COVID-19 as opposed to other potential confounding factors. This is a major limitation and should be mentioned.

Response 6. We appreciate this comment and acknowledge that a control group would improve the comparative aspect of our findings. While the inclusion of a healthy control group can provide important comparative data, it is not always feasible or necessary depending on the focus of the study. Healthy individuals can vary widely in terms of their baseline microbiota composition, immune responses, and lifestyle factors, making it challenging to effectively match study participants. We have added this shortcoming to the limitations of the study as follows.

"The study did not include a control group of uninfected individuals due to the variability in their baseline microbiota composition, immune responses, and lifestyle factors. Instead, we used relevant scientific literature as reference points to compare our results."

Please, see line 547-551.

7. Given the biological nature of the data, which often do not follow a normal distribution, verifying the assumptions of normality is crucial. Non-normally distributed data should ideally be analyzed using non-parametric tests such as the Wilcoxon signed-rank test.

Response 7: Thank you for your insightful comment. Recognizing that biological data frequently do not adhere to a normal distribution, we re-evaluated the CRP, IL-6, IL-1α, and IL-10 levels using the Wilcoxon signed-rank test. This non-parametric test is a suitable alternative to the paired t-test for data that do not satisfy normality assumptions. Consequently, we have revised the Results section to reflect the outcomes of this analysis and updated the legend of Figure 3 accordingly.

Please see to the revised manuscript on lines 367-393 for these changes.

8. With multiple tests being conducted, the risk of Type I errors (false positives) increases. The study does not mention any correction for multiple comparisons, such as the Bonferroni correction or False Discovery Rate (FDR) control.

Response 8: Thank you for this comment. In our study, the McNemar test was used to evaluate the presence or absence of various bacterial and fungal taxa over time. To address the risk of Type I errors due to multiple comparisons, we applied the Bonferroni correction using formula α=0.05/10. After applying this correction, only the p-values below 0,005 were considered statistically significant. The revised results are presented in Table 2.

Please also see the revised version in section Results, point 3.3.; line 264-310

9. Employing more comprehensive diversity metrics and community composition analyses, such as Principle Component Analysis (PCA) should be considered.

Response 9: Thank you very much for this valuable suggestion. Usually, PCA is useful to reduce large datasets into smaller components clustering together highly correlated variables while maintaining a significant proportion of the original information, even though this might introduce limitations when it comes to interpretability, especially in experimental data. While transforming the data, features inevitably lose the original meaning thus interpreting the components/factors in PCA may not be so straightforward. The other thing that might present problem with using the PCA is the fact that the PCA a data-driven method and is much sensitive to outliers, which in a modest sample size as ours might be an issue.

Another issue is that we didn’t analyse the entire spectrum of present microorganism but only those bacterial species identifying through MALDI-TOF mass spectrometry, so employing analyses like PCA might potentially result in biased interpretations of multivariate patterns, interactions and complex pathways between different species of microorganisms which all might add difficulties when interpreting the output of the PCA [https://doi.org/10.1093/sysbio/syv019].

Also, PCA assumes the linear relationship between variables, which in a limited dataset as ours might be hard to achieve. Finally, results from the PCA are usually much specific to the dataset used and often lack generality, as well as, they are often sensitive to experimental noise and bias. [https://doi.org/10.1186/1471-2105-16-S18-S7].

Given these limitations, we used alternative methodologies that were adapted to our data. Nevertheless, this is a great suggestion from the Reviewer and in our future work with the expansion of the microbiological analysis and hopefully adding more samples (combining data from other scientific partners) as well as additional data on the dietary consumption and patterns from food data and other lifestyle characteristics, we will be able to use this analysis and provide additional finding on the topic.

10. The study claims a significant change in gut microbiota composition during the initial phase of Omicron infection but reports no significant changes in overall diversity indices (Simpson’s and Shannon’s). This apparent contradiction needs further discussion and clarification.

Response 10. Thank you for your observation. We appreciate the opportunity to clarify the relationship between the observed changes in the gut microbiota composition and the reported diversity indices (Simpson’s and Shannon’s). Both Shannon's and Simpson's indices are measures of alpha diversity, reflecting both the richness (number of species) and evenness (distribution of species) within a community. While these indices provide a broad overview of diversity, they may not capture shifts in the composition of specific taxa unless these changes are substantial enough to affect the overall community structure. Our study did indeed find significant changes in the relative abundance of specific microbial taxa between the initial phase of Omicron infection and the post-recovery period. However, this does not necessarily translate into significant changes in the overall diversity indices like Simpson’s and Shannon’s.

As you suggested, we have expanded the Discussion section with a few sentences to better elaborate on this issue. Please see the revised Discussion section, line 423-430:

"Shannon’s and Simpson’s indices primarily assess the overall species richness and evenness within a community, which remained stable despite fluctuations in the relative abundance of individual taxa. The significant compositional changes observed indicate a dynamic response of the gut microbiota to Omicron infection, likely involving specific taxa rather than a broad alteration in community diversity. Therefore, while overall diversity metrics did not indicate significant differences, the underlying microbial community composition experienced notable alterations during the course of the disease."

11. There are several instances of typographical errors and inconsistent formatting throughout the manuscript that need to be corrected (e.g., inconsistent use of units and missing spaces).

Response 11. Thank you for this comment. We will thoroughly review the manuscript to correct typographical errors and ensure consistent formatting.

Round 2

Reviewer 1 Report

Comments and Suggestions for Authors

REVIEW

Dear authors,

The work presents a considerable improvement, the described methodology supports the results obtained, as well as mentioning the limitations of the research. However, there are still some details that need to be corrected.

Please consider the following comments to improve the content of your manuscript before publication. 

The determination of anti-SARS-CoV-2 IgG antibodies described in section 2.4 are not represented in section 3. Results, and therefore are not mentioned in section 4. Discussion, review and incorporate these results.

Lines 27, 438, 460, 469, 574: the acronym “spp.” it is not written in cursive.

Lines 114, 140: correctly write the volume unit “mL”.

Lines 269, 270: write the name of the microorganisms in cursive letters.

Line 310: write in cursive “Bacillus”.

Line 440: write in cursive “Enterococcus”.

Lines 489, 490: write in cursive “Bacillus and Lactobacillus”.

Please amend the requested comments and submit the revision file.

Author Response

Author's Response to Reviewer # 1

Dear authors, The work presents a considerable improvement, the described methodology supports the results obtained, as well as mentioning the limitations of the research. However, there are still some details that need to be corrected. Please consider the following comments to improve the content of your manuscript before publication. The determination of anti-SARS-CoV-2 IgG antibodies described in section 2.4 are not represented in section 3. Results, and therefore are not mentioned in section 4. Discussion, review and incorporate these results.

Response 1: Thank you for your insightful comment. We appreciate your attention to detail and agree that the anti-SARS-CoV-2 IgG antibodies should be discussed more thoroughly. We have reviewed our manuscript and have now incorporated the results regarding IgG antibodies into the Discussion section to provide a more comprehensive interpretation of our findings. Specifically, we added the following sentences: "Moreover, our findings showed that approximately 58% of participants had SARS-CoV-2 IgG antibodies at the beginning of the infection, with all participants showing the presence of IgG antibodies 30-35 days after symptom resolution. This seroconversion indicates a strong immune response, which may have contributed to the observed normalization of gut microbiota over time". Please, see line 512-516.

Comment 2:

Lines 27, 438, 460, 469, 574: the acronym “spp.” it is not written in cursive.

Lines 114, 140: correctly write the volume unit “mL”.

Lines 269, 270: write the name of the microorganisms in cursive letters.

Line 310: write in cursive “Bacillus”. Line 440: write in cursive “Enterococcus”.

Lines 489, 490: write in cursive “Bacillus and Lactobacillus”.

Response 2: Thank you for your detailed review and for pointing out the minor typographical errors in our manuscript. We have carefully addressed each of the issues you mentioned.

3. Please amend the requested comments and submit the revision file.

Thank you again for your valuable feedback, which has helped us improve our manuscript.

Reviewer 2 Report

Comments and Suggestions for Authors

The paper is much improved and may be accepted for publication.

Author Response

Thank you again for your support and valuable feedback, which has helped us improve our manuscript.

Reviewer 3 Report

Comments and Suggestions for Authors

1. The authors' reply that "The study did not include a control group of uninfected individuals due to the variability in their baseline microbiota composition, immune responses, and lifestyle factors. Instead, we used relevant scientific literature as reference points to compare our results" is befuddling and holds no water. Many high-quality studies on gut microbiota changes due to infections or diseases include control groups to strengthen their findings. Dealing with baseline variability is a common challenge in microbiome research. Researchers often use large, well-matched control groups or statistical methods to account for this variability. The justification for excluding a control group based on variability makes no sense and suggests a lack of methodological robustness.

2. Including a control group of individuals is feasible, particularly if they are matched for key variables such as age, gender, diet, and lifestyle.

3. The lack of significant findings after correction suggests the results may be due to random variation rather than real effects.

Comments on the Quality of English Language

Moderate edits needed.

Author Response

Author's Response to Reviewer # 3

Comment 1: The authors' reply that "The study did not include a control group of uninfected individuals due to the variability in their baseline microbiota composition, immune responses, and lifestyle factors. Instead, we used relevant scientific literature as reference points to compare our results" is befuddling and holds no water. Many high-quality studies on gut microbiota changes due to infections or diseases include control groups to strengthen their findings. Dealing with baseline variability is a common challenge in microbiome research. Researchers often use large, well-matched control groups or statistical methods to account for this variability. The justification for excluding a control group based on variability makes no sense and suggests a lack of methodological robustness.

Authors' Response 1: We appreciate the Reviewer’s detailed comments and suggestions.  In this response, we would like to provide additional explanations and rationale behind our decisions, as well as add further context and relevant high-quality reference studies to clarify our approach.

We acknowledge the fact that including a well-matched control group of uninfected individuals might enhance the robustness of a study when investigating potential associations. Our study design, on the other hand, focused on internal comparisons, i.e., analyzing changes in the gut microbiota and inflammatory markers in the same individuals over different time points (during the presence of the SARS-CoV-2 in the organism and 30-35 days after symptoms resolution). Many studies published in high-quality journals have used this or a similar approach. For example: The study "Gut Microbiota Composition is Associated with SARS-CoV-2 Vaccine Immunogenicity and Adverse Events" (Ng et al., 2022), published in the journal Gut journal (IF(2022) 24.5; https://doi.org/10.1136/gutjnl-2021-326563),, focused on intra-individual comparisons to assess changes in gut microbiota and immune responses before and after vaccination. This study did not include a separate control group of uninfected individuals. Instead, it collected stool and blood samples from participants at baseline (within three days of the first dose) and one month after the second dose of vaccination, focusing on changes within the same individuals over time. This is, in fact, very similar to the approach we used in our current study.

Additionally, the study "Gut Microbiota Dysbiosis Correlates with Abnormal Immune Response in Moderate COVID-19 Patients with Fever" (Zhou et al., 2021), published in the Journal of Inflammation Research (IF (2020) 6.922; https://doi.org/10.2147/JIR.S311518), also used internal comparisons within a specific patient population. This study compared gut microbiota composition and immune responses between moderate COVID-19 patients with and without fever, without including a separate control group of healthy individuals. Instead, it utilized the variability within the COVID-19 patient group to identify significant associations between gut microbiota and immune responses. Similarly, the study "Gut Microbiota Diversity and C-Reactive Protein Are Predictors of Disease Severity in COVID-19 Patients" (Moreira-Rosárioet al., 2021), published in Frontiers in Microbiology, (IF (2021) 6.064; https://doi.org/10.3389/fmicb.2021.705020), investigated the association between gut microbiota composition and COVID-19 severity in 115 patients. They categorized patients by disease severity and recovery location, focusing on internal comparisons without including a healthy control group. This study provides another example where internal comparisons within a patient population were utilized to draw significant conclusions about gut microbiota and disease severity without including an external control group of healthy individuals. All of these mentioned studies are well accepted by the scientific community and have provided valuable findings and conclusions.

As previously highlighted, our longitudinal approach helped to ensure the validity of our findings by comparing the gut microbiota composition at two time points within the same individuals. This allowed us to draw meaningful conclusions based on changes observed over time within the same person (when they were ill and when they were healthy). As demonstrated in previous studies, paired samples (pre/postevent samples) are the best approach when dealing with high variability.

Comment 2: Including a control group of individuals is feasible, particularly if they are matched for key variables such as age, gender, diet, and lifestyle.

Authors' Response 2: We agree that including a control group well-matched for key variables such as age, gender, diet, and lifestyle would be a good approach. However, this presented a significant challenge, as achieving a well-matched control group considering numerous variables like diet, lifestyle, comorbidities (type, severity, duration etc.), and ongoing therapies (type of medication, duration, dosage, regimen , etc.) was difficult or practically impossible considering our relatively small sample size. While matching individuals in relation to age and gender is relatively straightforward, matching with respect to all other (important) variables can be a very difficult and complex task, especially for a small scientific team with limited fundings, like ours.

Additionally, numerous studies demonstrate the significant role of dietary habits in shaping an individual's microbiota. For example, in one systematic review researchers clearly identified that “plant-based” and Mediterranean diets were associated with higher levels of Bacteroidetes and Firmicutes, while Western diets showed lower microbiota levels. Vegan/vegetarian diets correlated with increased Prevotella and decreased Clostridium. Variations in total energy intake were linked to changes in the abundance of Bacteroidetes and Firmicutes and increased microbial diversity. Also, fiber-rich diets were generally associated with increased microbial diversity and beneficial bacteria like Bifidobacterium and Lactobacillus. Resistant starch intake influenced bacterial proliferation, with notable effects on Eubacteria and Ruminococcus. Furthermore, higher protein intake was associated with less beneficial bacterial profiles. Please, see:

Diet-microbiota associations in gastrointestinal research: a systematic review. Duncanson, K. et al., Gut Microbes, 16(1) (2024), DOI: 10.1080/19490976.2024.2350785, https://www.tandfonline.com/doi/full/10.1080/19490976.2024.2350785 

In conclusion, neglecting these important variables and including inadequately matched controls, especially for variables/-factors that are insufficiently or incompletely known, could lead to misleading conclusions and potentially erroneous study findings. This is why we opted for an internal (paired-matched) comparison approach in our study. Finally, as specified in the study aim, our research focused entirely on gut microbiota changes in COVID-19 patients, during active disease and the period or (re)convalescence.

Comment 3: The lack of significant findings after correction suggests the results may be due to random variation rather than real effects.

Authors' Response 3: We thank the reviewer for advice in the first round of review to strengthen the rigor of the statistical analyses. Following these suggestion, we implemented more rigorous data analysis, including corrections for multiple comparisons, to mitigate the risk of Type I errors and validate our results.

Furthermore, our study design focused on internal comparisons, analyzing changes in gut microbiota and inflammatory markers within the same individuals over different time points. This longitudinal approach helped control of inter-individual variability and provided valuable insights into the dynamics of gut microbiota and immune response in COVID-19 patients. While the corrections for multiple comparisons reduced the number of significant findings, we believe this rigorous approach enhances the reliability and credibility of our results.

Also, we acknowledge the fact that our sample size was relatively modest, which might have led to the lack of significant findings after corrections for multiple testing. Thus, studies with a larger sample size and (possibly) longer period of follow up are warranted.

Considering the comments from the reviewer, we made following corrections in limitations of the study:

"The study did not include a control group of uninfected individuals due to the variability in their baseline microbiota composition, immune responses, and lifestyle factors. To validate our results, we applied rigorous statistical analyses, including corrections for multiple comparisons, to mitigate the risk of Type I errors. The longitudinal approach of analyzing the same individuals at two time points allowed us to control for inter-individual variability".